# Contrasting activation energies of litter-associated respiration and P uptake drive lower cumulative P uptake at higher temperatures

Nathan J. Tomczyk[1], Amy D. Rosemond[1], Anna Kaz[2], and Jonathan P. Benstead[3].

1Odum School of Ecology, University of Georgia, Athens, Georgia 30606 USA

5    2Department of Oceanography and Coastal Sciences, Baton Rouge, Louisiana 70803 USA

3Department of Biological Sciences, University of Alabama, Tuscaloosa, Alabama 35487 USA

*Correspondence to*: Nathan Tomczyk (Nathan.tomczyk@gmail.com)

**Abstract.** Heterotrophic microbes play key roles in regulating fluxes of energy and nutrients, which are increasingly affected by globally changing environmental conditions such as warming and nutrient enrichment. While the effects of temperature 10    and nutrients on microbial mineralization of carbon have been studied in some detail, much less attention has been given to how these factors are altering uptake rates of nutrients. We used laboratory experiments to simultaneously evaluate the temperature dependence of soluble reactive phosphorus (SRP) uptake and respiration by leaf litter-associated microbial communities from temperate headwater streams. Additionally, we evaluated the influence of the initial concentration of SRP on the temperature dependence of P uptake. Finally, we used simple simulation models to extrapolate our results and estimate 15    the effect of warming and P availability on cumulative gross uptake. We found that the temperature dependence of P uptake was lower than that of respiration (0.48 vs. 1.02 eV). Further, the temperature dependence of P uptake increased with the initial concentration of SRP supplied, ranging from 0.12 to 0.48 eV over an 11 to 212 µg L$^{-1}$ gradient in initial SRP concentration. Finally, despite our laboratory experiments showing increases in mass-specific rates of gross P uptake with temperature, our simulation models predict declines in cumulative P uptake with warming, because the increased rates of respiration at warmer 20    temperatures more rapidly depleted benthic carbon substrates and consequently reduced the biomass of the benthic microbial community. Thus, even though mass-specific rates of P uptake were higher at the warmer temperatures, cumulative P uptake was lower over the residence time of a pulsed input of organic carbon. Our results highlight the need to consider the combined effects of warming, nutrient availability, and resource availability/magnitude on carbon processing as important controls of nutrient processing in heterotrophic ecosystems.

## 1 Introduction

Microbial communities regulate ecosystem nutrient cycling and retention through their uptake and mineralization of nutrients (Burgin et al., 2011; Brookshire et al., 2011). Thus, any environmental factor that affects cell nutrient quotas, biomass,

or production of microorganisms can influence rates of ecosystem nutrient processing (Cross et al., 2005, 2015). Notably, increases in nutrient concentrations and temperature are both expected to increase rates of microbial metabolism and growth (Brown et al., 2004; Sterner and Elser, 2002), and such increases are being observed across human-influenced landscapes (Kaushal et al., 2010; Stets et al., 2020). Any increase in microbial community metabolism should be associated with higher demand for nutrients, as measured by gross nutrient uptake at the ecosystem level (Hall and Tank, 2003). In autotrophic systems, increases in temperature drive increases in gross primary production, resulting in predictably higher demand for nutrients (Rasmussen et al., 2011); however, in donor-controlled detrital systems, such as soils and forest streams, increased rates of metabolism stimulated by increases in temperature or nutrients can lead to reductions in pools of the dead organic matter that fuels metabolism, eventually reducing microbial biomass on an areal basis(Walker et al., 2018; Suberkropp et al., 2010). Thus, long-term responses of nutrient uptake to higher temperatures and nutrient concentrations are challenging to parse in detritus-dominated ecosystems, as mass-specific rates of uptake may increase even as total microbial biomass declines, complicating net ecosystem responses.

Mechanisms explaining the joint effects of temperature and nutrients on mass-specific rates of nutrient uptake ($U$) remain poorly resolved. Temperature may cause increases in nutrient demand that directly match increases in metabolism (Allen and Gillooly, 2009). Alternatively, increases in nutrient demand may deviate from metabolism for two reasons. First, temperature may influence the nutrient use efficiency of microbes. For example, algae can use nutrients more efficiently at higher temperatures, expressed as an increase in the ratio of carbon (C) to nitrogen (N) or phosphorus (P) in their biomass (Thrane et al., 2017; De Senerpont Domis et al., 2014; Yvon-Durocher et al., 2015). Bacteria and fungi can also exhibit variation in their demand for nutrients relative to their carbon demand (Gulis et al., 2017; Scott et al., 2012), but it is unknown whether their biomass stoichiometry varies systematically with temperature (Cross et al., 2015). If bacteria and fungi also increase their nutrient use efficiency in response to rising temperatures, temperature may increase metabolism and respiration more than $U$ (Hood et al., 2018). Second, basal metabolic costs may increase with warming. As a consequence of increased basal metabolic costs, the carbon use efficiency (biomass produced relative to carbon assimilated) of heterotrophic microbes may decline with increasing temperature (Manzoni et al., 2012; Li et al., 2019; Doi et al., 2010). Decreased carbon use efficiency implies an increase in carbon use relative to nutrient demand if stoichiometry remains fixed. Despite differences in mechanism, both declines in carbon use efficiency and increases in nutrient use efficiency imply a greater increase in demand for carbon than for nutrients at higher temperatures.

Responses of nutrient uptake to higher nutrient concentrations are also potentially complex. Uptake of nutrients is often limited by the concentration of dissolved nutrients (Mulholland et al. 2008). As nutrient concentrations increase, uptake rates typically increases to a plateau (Dodds et al., 2002). At low nutrient concentrations, uptake is generally limited by the encounter rate between nutrient molecules and cell membranes; at high concentrations, uptake is instead limited by the rate of transfer of nutrients across cell membranes. These dynamics are generally described by Michaelis-Menten kinetics

(Weigelhofer et al., 2018). Consequently, the proportion of dissolved nutrients taken up by the microbial community may decline with increasing nutrient concentration (O'Brien et al., 2007). Organismal measurements of nutrient use efficiency have also demonstrated that increasing nutrient supply relative to carbon leads to less efficient use of nutrients, as demonstrated by lower biomass C:nutrient content (Godwin and Cotner, 2015). Lastly, the combined effects of nutrients and temperature on $U$ may be additive or nutrient concentration and temperature may interact to determine growth and uptake rates (Cross et al., 2015; Davidson et al., 2012). However, to date there is little evidence that the effects of nutrients and temperature are strongly interactive, at least in detritus-based systems (Manning et al., 2018).

Regardless of temperature and nutrient availability, ecosystem-level nutrient uptake is also a function of substrate availability and total microbial biomass. Much of the benthic metabolism in forest streams and soils is supported by inputs of allochthonous organic matter, and particularly leaf litter from the terrestrial environment (Tank et al., 2018; Wallace et al., 2015). In temperate ecosystems with deciduous vegetation, there is strong seasonality in the input of senescent leaf litter. This finite supply of litter is subsequently depleted by the activity of microbial and animal consumers (Wallace et al., 2015; Webster and Tank, 2000; Marks, 2019). While increased temperature and nutrients stimulate mass-specific rates of metabolism, they also accelerate the loss of benthic carbon, which eventually reduces microbial biomass at the ecosystem level (Walker et al., 2018; Suberkropp et al., 2010). The importance of these dynamics for rates of ecosystem nutrient uptake and metabolism have been illustrated empirically; studies have found an apparent negative effect of temperature on nutrient uptake that is partially driven by seasonal changes in microbial biomass in forest streams (Hoellein et al., 2007; Valett et al., 2008), which tends to peak in the winter after leaf litter inputs have entered the stream, and then decline in the summer as the pulse of detrital carbon is depleted (Suberkropp et al., 2010). While these studies have illustrated the importance of carbon standing stocks as a control of ecosystem nutrient uptake, the consequences of increased temperature and nutrient concentration for cumulative nutrient uptake remain unexplored. Because the seasonal supply of carbon in forest stream ecosystems is finite within an annual cycle, the cumulative amount of nutrient uptake over the residence time of the detritus is important to consider, though challenging to evaluate empirically.

Here, we quantify how stream temperature and the concentration of soluble reactive P (SRP) affect gross uptake of P ($U_{srp}$) by leaf litter-associated microorganisms in forested headwater streams and evaluate whether increases in $U_{srp}$ match warming-induced increases in metabolic rates (measured as respiration). We hypothesized that higher temperatures would drive increased respiration rates and $U_{srp}$, though we expected that $U_{srp}$ would increase less with temperature than respiration due to changes in carbon or nutrient use efficiency. We also hypothesized that the temperature dependence of $U_{srp}$ would vary based on the concentration SRP supplied, with low concentrations of SRP constraining the temperature dependence of $U_{srp}$ (Cross et al. 2015). Further, we hypothesized that temperature would modify relationships between nutrient concentration and $U_{srp}$. Specifically, we expected that higher temperatures would increase maximum uptake rates while decreasing the half-saturation constants of Michaelis-Menten models (Cross et al. 2015). To test these hypotheses, we quantified the temperature

dependence of $U_{srp}$ in laboratory experiments, tested whether this temperature dependence varied across nutrient concentrations, and compared it to the temperature dependence of respiration. Finally, we hypothesized that, if the temperature

dependence of respiration is greater than that of $U_{srp}$, the consequence would be a reduction in cumulative $U_{srp}$ over the residence time of a pulsed leaf litter input, caused by faster loss of leaf-associated carbon at higher temperatures. To test this, we used simple simulation models to extrapolate our measured effects of temperature on carbon processing and $U_{srp}$ and quantified the effect of warming on cumulative $U_{srp}$ over the residence time of a single seasonal input of leaf litter.

## 2 Methods

### 2.1 Comparing the temperature dependences of SRP uptake and respiration

We conditioned leaf litter for these experiments at the United States Department of Agriculture Forest Service Southern Research Station Coweeta Hydrologic Laboratory (CHL) in the southern Appalachian Mountains, Macon County, North Carolina, USA (see Swank and Crossley [1988] for site information). We incubated *Rhododendron maximum* (hereafter, *Rhododendron*) leaf litter to allow for microbial colonization in Watershed 5a in 5-mm mesh litterbags for 114 days beginning

on 17 November 2020. We removed a subset of the bags on 11 March 2021 and returned them to the laboratory, where we cut the leaves into fragments approximately 1.5cm × 1.5cm. We placed these fragments in 1-L bottles full of aerated stream water, which we incubated in water baths at five different temperatures (4, 8, 12, 16, 20°C). Each water bath contained three bottles, which we consider replicates, though we acknowledge the bottles are not fully independent. After we acclimated the microbial communities for 24 h, we removed leaf fragments from the bottles to measure either their gross SRP uptake or respiration rate

(see below). We repeated this procedure (only the 4°C to 16°C temperature treatments) on 18 March 2021 and pooled the results for analysis.

We used three subsamples from each replicate bottle to measure respiration rates. To estimate respiration rates, we filled 20-ml scintillation vials with stream water at the appropriate treatment temperature and measured the initial concentration of oxygen using a YSI 5100 Dissolved Oxygen Meter (YSI Inc, OH, USA). After measuring initial concentrations of oxygen,

we immediately replaced the water that was displaced during the initial measurement (~15% of the volume) with stream water from the same bottle that was initially used to fill the vial. Then, we added several leaf fragments (similar amounts among vials) to the vial and secured the cap such that no air remained in the vial. We prepared three blanks (water but no leaves added) along with the samples in each temperature treatment. We then returned the vials to the water bath to incubate in the dark for 2 to 7 h, giving the vials in colder temperatures more time to incubate to ensure meaningful changes in the

concentration of dissolved oxygen. After incubation, we recorded the final concentration of dissolved oxygen, removed the leaves, dried them to a constant mass, and weighed them. We calculated respiration rates (mg $O_2$ hr$^{-1}$ mg$^{-1}$) based on the difference in the mass of oxygen in the vial before ($O_{2\text{-pre}}$, mg) and after the incubation ($O_{2\text{-post}}$, mg), minus the change in

oxygen in the blanks ($O_{2\text{-pre-blank}}$ and $O_{2\text{-post-blank}}$ ), divided by incubation time (T, h) and the dry mass of leaves in the vial (M, g, equation 1).

$$Respiration\ rate = \frac{(O_{2-pre}-O_{2-post})-(O_{2-pre-blank}-O_{2-post-blank})}{T*M} \qquad ( \qquad\qquad (Equ.\ 1)$$

We also used three different subsamples from each replicate bottle to measure rates of $U_{srp}$ simultaneously with the measurements of respiration. We amended reservoirs of stream water at each temperature with nutrients to elevate concentrations from <5 ug L$^{-1}$ to ~30 to 60 µg L$^{-1}$ SRP. We then dispensed 40 mL of this nutrient-amended water into 50-mL centrifuge tubes, and added several leaf fragments. Three blanks (i.e., water with no leaves added) were prepared along with
each temperature treatment. After 2 to 7 hours of incubation, we removed a subsample of the water with a syringe and filtered it through an AE-grade glass fiber filter (nominal pore size 1.0-µm, Sterilitech, WA, USA), and immediately froze the sample for preservation. We determined SRP concentrations using an Alpkem Rapid Flow Analyzer 300 (Alpkem, College Station, Texas, USA). We retained leaf fragments, dried them to a constant mass, and weighed them. We calculated $U_{srp}$ as the difference in the mass of SRP between the mean of the blanks (P$_{blank}$, µg P) and each subsample (P$_{sample}$, µg P), normalized to
the dry mass of leaves and the incubation time (equation 2).

$$U_{srp} = \frac{P_{sample}-P_{blank}}{T*M} \qquad\qquad (Equ.\ 2)$$

To quantify the effects of temperature on rates of respiration and $U_{srp,}$ we estimated their activation energies ($E_a$) using the Boltzmann-Arrhenius equation (equation 3, Brown et al., 2004), where the rate of the process ($r_i$) is a function of the rate at a reference temperature ($r_{ref}$), the activation energy ($E_a$), the temperature in kelvin ($T$), and the Boltzmann constant ($k_B$;
8.617×10$^{-5}$ eV K$^{-1}$). We averaged the subsample measurements from each bottle and fit our data to the linearized version of the Boltzmann-Arrhenius equation, with temperature centered on a standard temperature ($T_{12}$, 12ºC), by regressing the log$_e$-

transformed process rates against the standardized Boltzmann temperature (equation 3), and estimating the $E_a$ based on the slope of this line (equation 4).

$$r_i = r_{ref} * e^{\frac{-Ea}{k_B*T}},$$ (Equ. 3)

$$\ln(r_i) = \ln(r_{12}) + \frac{1}{k_B*T_{12}-k_B*T} * -E_a,$$ (Equ. 4)

The two dates on which the experiment was run may have had different biological or environmental conditions, so we included a categorical effect of date in our statistical models to account for any differences. To evaluate whether the responses of respiration and $U_{SRP}$ to temperature were different we used an ANCOVA-type linear model. To do this, we fit a linear model that described the $\log_e$-transformed rates of respiration and uptake rates as a function of the standardized Boltzmann temperature. The model included an interaction between temperature and a binary variable that indicated the type of rate (i.e., respiration or uptake). A significant interaction term in this model indicates that the slopes of the relationships between temperature and these rates differ. Finally, as an alternative way to evaluate relative differences in metabolism and P demand, we converted mass-based units of $O_2$ and SRP to their molar equivalents, and converted oxygen to units of C assuming a respiratory quotient of 0.85 (moles $CO_2$ produced per mole $O_2$ consumed; Bott 2006). Then, we calculated the molar ratio of C respired to $U_{srp}$, which we report as the C:P of respiration to uptake. We tested the effects of temperature on the $\log_e$-transformed molar ratio, using the centered inverse Boltzmann temperature as the predictor variable.

## 2.2 Effect of nutrient concentration on temperature dependence of SRP uptake

We conducted a separate experiment to test whether the initial concentration of nutrients affected the temperature dependence of nutrient uptake. We incubated *Acer rubrum* (hereafter, *Acer*) leaves in Lower Hugh White Creek at the CHL for approximately 30 d during summer 2019 and then returned the leaves to the laboratory. We used a shorter incubation time for *Acer* than for *Rhododendron* due to higher environmental temperatures and generally more rapid colonization of this more labile litter species. We added several whole leaves to 250-mL Nalgene bottles with 200 mL water and incubated them for approximately 3 h. Leaves were incubated at six temperatures ranging from 4°C to 21°C and eight initial SRP concentrations ranging from 11 to 217 µg L$^{-1}$ that were created by adding a concentrated solution of $KH_2PO_4$ to the stream water. After incubation, we removed a subsample of water with a syringe, filtered it, and froze it immediately to preserve the sample. We then analyzed the water samples for SRP using a spectrophotometer (Shimadzu UV-1700) and the ascorbic acid method (APHA, 1995). Each temperature and concentration combination had two replicates and one blank that did not have leaves added. Leaf fragments incubated in each bottle were dried and weighed after the incubations as above.

We calculated $U_{srp}$ in the same manner described above (equation 2). We then used two techniques to quantify how the initial concentration of nutrients and temperature interacted to affect rates of $U_{srp}$. First, we grouped the data based on the initial concentration and estimated the temperature dependence of $U_{srp}$ at each initial nutrient concentration. We estimated the effect of temperature using the linearized version of the Boltzmann-Arrhenius equation (equation 4), by regressing the $\log_e$-transformed $U_{srp}$ rates against the centered inverse Boltzmann temperature, and estimated the $E_a$ based on the slope of this line.

Then, we evaluated the effect of the initial concentration of SRP on the temperature dependence of $U_{srp}$ by estimating the slope of the relationship between initial SRP concentration and the activation energy of $U_{srp}$ at each concentration, evaluating both a linear and saturation response of the activation energy of $U_{srp}$ to temperature. In a second analysis of the same data, we grouped the data by temperature and estimated the effect of changes in initial nutrient concentration at different temperatures. We fit models of Michaelis-Menten kinetics to nutrient concentration and $U_{srp}$ at each temperature, in which we modeled $U_{srp}$ as a

function of initial SRP concentration [SRP] and two parameters, the maximum uptake rate ($U_{max}$) and the half-saturation constant ($k_m$, equation 5):

$$U_{srp} = \frac{[SRP] * U_{max}}{[SRP] + k_m} \qquad \text{(Equ. 5)}$$

We then evaluated the influence of temperature on the Michaelis-Menten parameters using the framework of metabolic theory. We regressed $\log_e$-transformed values of $k_m$ and $U_{max}$ against the standardized Boltzmann temperature to estimate the activation

energy of each of these parameters.

2.3 **Simulating the direct and indirect effects of temperature and enrichment on SRP uptake**

      We used a simple simulation model to evaluate how temperature and SRP concentration affect cumulative $U_{srp}$ over the residence time of a pulsed leaf litter input. These simulations consider both the direct effects of SRP concentration and

temperature on mass-specific $U_{srp}$ and the indirect effects mediated through depletion of litter-associated carbon. These simulations were designed to illustrate the dynamic consequences of our laboratory measurements and inform a more comprehensive representation of carbon and nutrient cycles in forested streams. The simulated stream reach starts with 315 g leaf C m$^{-2}$, which is based on observations of leaf standing stocks in streams at CHL (Suberkropp et al., 2010). Mass-specific rates of leaf mass loss were estimated as a function of temperature and, in some scenarios, nutrient concentration (see below

for details on scenarios). We estimated mass-specific rates of $U_{srp}$ as a function of temperature and SRP concentration using data from our experiments or from the literature (see below). We then calculated areal rates of gross SRP uptake as the product of mass-specific $U_{srp}$ and the areal mass of C remaining in the stream. For both rates of $U_{srp}$ and respiration we converted mass-specific rates from units of dry-mass to units of carbon assuming an average leaf carbon content of 45%. We report cumulative $U_{srp}$, when 99% of the leaves were consumed by microbial metabolism.

We considered the effects of warming and nutrient enrichment on cumulative $U_{srp}$ in four scenarios (Table 1); in each scenario we evaluated the effect of warming using a low temperature of 10°C and a high temperature of 14°C. First, we considered the effect of warming on cumulative $U_{srp}$ when both respiration and uptake have the same temperature dependence

of 0.65 eV (Brown et al., 2004). In this model, we used estimates of $r_{ref}$ of respiration from our first experiment and $r_{ref}$ of $U_{srp}$ from our 19 µg L$^{-1}$ treatment (i.e., a low-to-moderate concentration). Second, we simulated $U_{srp}$ and respiration using our

measured temperature dependence values, using the temperature dependence of respiration from our first experiment and the measured temperature dependence of $U_{srp}$ from the 19 µg L$^{-1}$ treatment in our second experiment. Third, we simulated uptake at a higher nutrient concentration, using the temperature dependence of $U_{srp}$ from the 111 µg L$^{-1}$ treatment in our second experiment and the temperature dependence of respiration from our first experiment. Fourth, we simulated uptake with our estimates of $U_{srp}$ at the high concentration of 111 µg L$^{-1}$, and included a factor to account for the effect of nutrient enrichment

on respiration of 1.32× (Manning et al., 2018). We propagate uncertainty in our parameter estimates of temperature dependences by bootstrapping our estimates of cumulative $U_{srp}$ 1000 times and compare outcomes of the simulations to estimate effect sizes. We do not include statistical analysis of the outcomes of these simulations.

## 3 Results

### 3.1 Comparing the temperature dependences of SRP uptake and respiration

We estimated an $E_a$ of respiration during the laboratory experiment of 1.02 eV (SE 0.06), which is higher than the established canonical value for cellular respiration (0.60 - 0.70 eV; Brown et al. 2004, Figure 1a). We estimated an $E_a$ of $U_{srp}$ of 0.48 eV (SE 0.05), which was significantly lower than the $E_a$ of respiration (estimated difference in $E_a$ = 0.48, SE 0.09, $F_{1,48}$ = 28.22, $P$ <0.0001, Figure 1a, b). Thus, there was a significant increase in the ratio of carbon respired relative to $U_{srp}$ (Figure

1c, Table 2), which increased with an $E_a$ of 0.54 eV (SE 0.08). In back-transformed units, this effect roughly translates to an increase in the C:P of respiration to uptake of 2.54 moles of C per mole of P with a one-degree increase in temperature (Figure 1c).

### 3.2 Effect of nutrient concentration on temperature dependence of SRP uptake

Temperature and the initial concentration of SRP both played an important role in determining rates of nutrient uptake

(Figure 2). The $E_a$ of $U_{srp}$ was greater at higher concentrations (Figure 2, Table 2), though the rate of increase was greater at low concentrations and saturated at higher nutrient concentrations (Figure 2). We found that temperature influenced patterns of SRP uptake across nutrient concentrations (Figure 3, Table 2). Temperature increased $U_{max}$, with an $E_a$ of 0.55 eV (SE 0.16) but did not have a detectable effect on $k_m$ (Figure 3).

### 3.3 Simulating the direct and indirect effects of temperature and enrichment on SRP uptake

Across all simulations, warmer temperatures consistently reduced cumulative $U_{srp}$ (Figure 4a). The reductions in cumulative $U_{srp}$ were a direct consequence of the accelerated loss of leaf-associated carbon, which outweighed the effect of increased mass-specific rates of $U_{srp}$ later in the simulations (Figure 4b). While warming reduced cumulative $U_{srp}$ in each simulation, the magnitude of the reduction depended on both the nutrient concentration and the temperature dependence parameters we used to simulate mass-specific rates of $U_{srp}$ and respiration. Our simulations that had the same activation energy

for both respiration and $U_{srp}$, projected 0.81× the cumulative $U_{srp}$ in the warm scenario (14ºC) compared to the cold scenario (10ºC) (Figure 4a). However, when we simulated these processes using measured activation energies of respiration and $U_{srp}$ measured at the low SRP concentration, we found that the effect of warming was greater, with cumulative $U_{srp}$ in the warm scenario equal to 0.62× that in the cool scenario (Figure 4a). At the higher SRP concentration, the absolute effect of warming on cumulative P uptake was greater than at the lower concentration (i.e., absolute differences of 4.7 vs. 12.0 g SRP m$^{-2}$, Figure

4a). However, the relative effect of warming on P uptake was smaller at the higher nutrient concentration, with cumulative uptake in the warm scenario 24% lower than in the cool scenario regardless of the effect of enrichment on respiration (Figure 4a).

Increases in nutrient concentration increased cumulative $U_{srp}$ in our simulations. In the cool scenario, cumulative $U_{srp}$ was 4× higher at the higher SRP concentration (Figure 4a). Similarly, in the warm scenario, cumulative $U_{srp}$ was 4.9× higher

at the high compared to low nutrient concentration (Figure 4a). These differences in cumulative $U_{srp}$ due to differences in concentration were somewhat smaller when we included the effect of nutrient enrichment on respiration, falling to 3× and 3.7×, respectively, at the low and high temperature (Figure 4a).

## 4 Discussion

We observed a lower activation energy for $U_{srp}$ than for respiration in our experiment, indicating the potential for shifts in carbon and nutrient processing as temperatures increase in forest streams. Additionally, we found that the temperature dependence of $U_{srp}$ increased as the concentration of SRP supplied increased. Simulated estimates of cumulative $U_{srp}$ highlighted that, even though temperature increased instantaneous rates of $U_{srp}$, the indirect effect of temperature on benthic carbon standing stocks led to lower cumulative $U_{srp}$ at higher temperatures. Together our results highlight that warming will

likely alter rates of gross nutrient uptake in forest streams, but the magnitude and direction of these effects may depend on the spatial and temporal scale of interest, as well as the nature of the carbon resources available.

Our finding that $U_{srp}$ increased less with temperature than microbial metabolism is in concordance with findings from other ecosystems. In a field experiment, Hood et al. (2018) found that stream warming increased primary production almost three-fold, while it had no measurable effect on rates of nutrient uptake. This was attributed to increases in the efficiency of

nutrient recycling, mineralization, and N$_2$ fixation (Hood et al., 2018). In our leaf-microbe system, factors such as increased nutrient recycling, or an increasing proportion of nutrient demand being satisfied through "mining" of leaf nutrients, may explain the reduced temperature dependence of $U_{srp}$ that we observed. Additionally, some of the increased respiration with temperature may be due to an increase in basal metabolic costs, which would not require a matched increase in nutrient demand (Manzoni et al., 2012; Li et al., 2019; Doi et al., 2010). Nutrient demands of heterotrophs may also shift with higher

temperatures. In our litter-microbe system, fungi are of particular interest as they dominate leaf microbial communities (Findlay et al., 2002). A previous study found that the elemental content of *Agaricomycetes* fruiting bodies was correlated with environmental temperature, with temperature increasing their biomass C:P (Zhang and Elser, 2017). Additionally, the ratio of

C respired through respiration to P taken up in this study is low relative to the mean biomass C:P of fungi and bacteria (Godwin and Cotner, 2014; Zhang and Elser, 2017). This likely indicates some luxury uptake of SRP, which is known to occur at the ecosystem scale (Payn et al., 2005) and within fungal tissue (Gulis et al., 2017) when nutrient concentrations are temporarily elevated.

Nominally, our finding that temperature increases mass-specific $U_{srp}$ is counter to previous examinations of the effect of temperature on rates of nutrient uptake in forest streams (Hoellein et al., 2007; Valett et al., 2008). However, while these previous studies found negative effects of temperature on nutrient uptake, their results highlighted the dominant role of microbial biomass as a control on nutrient uptake. In forest streams, biomass of the microbial community is tightly linked to the standing stock of detrital carbon, which varies inversely with seasonal temperatures in temperate forest streams (Hoellein et al., 2007; Valett et al., 2008; Suberkropp et al., 2010). While these studies potentially illustrate the role of heterotrophic microbial biomass in nutrient uptake, the observed winter peaks in nutrient uptake in these studies may be driven in part by increased autotrophic production allowed by a relatively open canopy during winter. The importance of both direct physiological and indirect biomass-mediated effects of temperature on ecosystem processes has been appreciated in detritus-based systems (e.g., Wilmot et al., 2021). However, our study is the first to separate the contribution of these two processes to patterns of long-term cumulative nutrient uptake. Specifically, when we considered only the direct effect of temperature on mass-specific rates of $U_{srp}$, we inferred that $U_{srp}$ increases with temperature. However, when we incorporated the effect of temperature on respiration we found that this indirect effect of warming decreased cumulative $U_{srp}$ (Figure 4a).

The aim of our simple simulation models was to isolate the dynamic consequences that our experimental results imply and explore their relevant long-term outcomes. As such, we ignored the dynamic process of biomass accumulation on leaves (Gulis et al., 2008), which is affected by both temperature and nutrients. Additionally, our simulations consider microbial respiration as the only mechanism of leaf mass loss. Under natural conditions, microbial fragmentation, physical abrasion, and consumption by macroinvertebrates can all drive meaningful amounts of leaf breakdown (Marks, 2019; Wilmot et al., 2021). Furthermore, our simulations were conducted at a constant temperature, which would lead to depressed rates of breakdown relative to simulations that include temperature variability (Tomczyk et al., 2020). Not including these processes in our model likely explains the high residence time of leaves in our simulations; at the low nutrient concentration and temperature our simulations had leaf residence times over 1000 days (Figure 4b), while field studies have found residence times of around two years for *Rhododendron* leaves in relatively pristine streams (Manning et al., 2015). Increasing rates of breakdown in our simulations to mimic residence times observed in the field leads to cumulative uptake in the cool scenario exceeding cumulative uptake in the warm scenario after only 139 days (Appendix A). Furthermore, while our models did not include seasonal inputs of leaves, our general finding that cumulative uptake in cool scenarios is greater than in warm scenarios is robust to successive seasonal inputs of leaves (Appendix B). Our treatment of the effect of nutrients on respiration is fairly simple, and comes from a study in which water was amended with both N and P, not just P as we consider throughout this study (Manning et al., 2018). While our simulation models do not incorporate all the complexity of stream ecosystems, the

consequence of the differences in the temperature dependence of carbon and nutrient cycle processes should persist in both more complex ecosystem models and in the natural environment.

Much like the carbon cycle processes of gross primary production, ecosystem respiration, and net ecosystem production, nutrient cycles in streams are comprised of positive and negative gross fluxes, the balance of which dictates net nutrient exchange between the water column and benthos (von Schiller et al., 2015; Brookshire et al., 2009). While we focus exclusively on the gross flux of nutrients from the water column to the benthos (i.e., $U$) in this analysis, relationships between temperature, gross nutrient release, and net nutrient exchange should also be examined to understand how nutrient cycling will change with warming. One detailed simulation of stream nutrient dynamics, which included the same temperature dependence for gross nutrient uptake and mineralization, predicted warming would cause declines in the net uptake of both N and P ranging from 0.9-4.3% (Webster et al., 2016). These modeled declines in net nutrient exchange were driven by the faster mineralization of organic matter that occurs at warmer temperature. Similarly, net mineralization of nutrients has been observed at higher temperatures in terrestrial systems; in one study, increased watershed nitrate export was linked to warming-induced increases in soil N mineralization (Brookshire et al., 2011). Periods of warming have also been linked to increases in *net* N mineralization and nitrate accumulation in agricultural soils (Liang et al., 2011), while experimental warming increased nitrate leaching in tundra soils (Harms et al., 2019). Thus, while our study focuses exclusively on $U$ as a gross flux, mineralization and net nutrient exchange are important aspects of stream nutrient cycling that are also likely temperature dependent.

While much research has addressed the effects of warming on carbon cycle processes (Davidson and Janssens, 2006; Song et al., 2018), far less attention has been paid to how warming affects nutrient cycles, despite the importance of these processes for ecosystem function (Peterson, 2001; Conley et al., 2009). Much of the interest in the effects of temperature on nutrient cycling has been at the level of the individual organism, including surveys of the effects of temperature on organismal stoichiometry (Yvon-Durocher et al., 2015; Zhang and Elser, 2017; Woods et al., 2003; Yuan and Chen, 2015). While some studies have shed light on ecosystem-level changes in nutrient cycling caused by temperature (Brookshire et al. 2011, Liang et al. 2011, Hood et al. 2018, Harms et al. 2019), more work is needed to reveal the underlying mechanisms of temperature effects on carbon-nutrient interactions. The results of our study, although only a small step, highlight that nutrient uptake is dependent on temperature but uncoupled to increases in carbon demand, and that the direction of the effect of warming on nutrient uptake is sensitive to the time scale that is considered (i.e., instantaneous vs. over months or years).

## 5 Conclusions

In this study we compared the effects of temperature on rates of respiration and $U_{srp}$ by leaf-associated microbial communities, as well as how SRP concentration altered the relationship between temperature and $U_{srp}$. Experimental changes in temperature increased mass-specific rates of both respiration and $U_{srp}$, though the increases in $U_{srp}$ were smaller than increases in respiration. The relationship between temperature and $U_{srp}$ changed with the concentration of SRP supplied, and the response to temperature was greater at high nutrient concentrations. However, despite the fact that our experiment found increases in mass-specific rates of $U_{srp}$ with temperature, our simulation models predicted declines in cumulative $U_{srp}$ over the

residence time of a single seasonal input of leaves, primarily as a consequence of more rapid depletion of leaf litter. The relative magnitude of this decrease may be greater in oligotrophic systems where increases in mass-specific $U_{srp}$ are more constrained.

Microbial metabolism and nutrient processing are being altered by climate change (Brookshire et al., 2011; Song et al., 2018). This study highlights that changes in rates of metabolism may not perfectly predict changes in rates of gross nutrient demand, as simple stoichiometric models may predict (Cross et al., 2015). While our study highlights differences in the response of respiration and $U_{srp}$ to temperature, further research is required to understand the cause of this divergence in process rates; we suspect changes in nutrient use efficiency and/ or carbon use efficiency with temperature drive the patterns we observed. Furthermore, our study highlights the dominant role that carbon supply plays in determining rates of nutrient cycling in detrital systems (Valett et al., 2008). Understanding general relationships between warming and nutrient cycling, with a particular consideration for the interconnectedness of the carbon and nutrient cycles (Schlesinger et al., 2011), will be important for understanding the future of nutrient cycling, including patterns of export from warming ecosystems.

*Code and data availability:* All data and result are included in the online repository for this paper: https://github.com/nathantomczyk/Temperature-Nutrient-Uptake

*Acknowledgements:* This study received support from the National Science Foundation (DEB-1655789 to ADR and DEB-1655956 to JPB). This work was also supported by a small grant from the Odum School of Ecology at the University of Georgia. This manuscript benefited greatly from comments from Carolyn Cummins, Phillip Bumpers, Laura Naslund, Craig Osenberg, Seth Wenger, Vlad Gulis, Danielle Hare, and Erin Hotchkiss.

*Author Contributions:* JBNJT, JPB, AK, and ADR developed the ideas for this manuscript. Data were collected by NJT and AK. NJT led the data analysis and wrote the first draft, and AK, ADR, and JPB contributed to revisions of the manuscript.

*Competing interests:* No competing interests.

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

Table 1: Summary of parameters used in different scenarios of simulations models. Reference rates of processes ($r_{ref}$) are
presented in units of mass of oxygen per gram of leaf ash-free dry mass per hour at 12°C. The temperature dependence of
processes is represented as the activation energy ($E_a$).

| | Scenario 1 | Scenario 2 | Scenario 3 | Scenario 4 |
|---|---|---|---|---|
| Title on graph | Low P $E_a$ =0.65 | Low P measured $E_a$ | High P measured $E_a$ | High P measured $E_a$ P effect on respiration |
| $r_{ref}$ of respiration (mg $0_2$ g$^{-1}$ hr$^{-1}$) | 0.04 | 0.04 | 0.04 | 0.04 |
| $r_{ref}$ of SRP uptake (µg P g$^{-1}$ hr$^{-1}$) | 1.94 | 1.94 | 8.58 | 8.58 |
| $Ea$ of respiration (eV) | 0.65 | 1.02 | 1.02 | 1.02 |
| $Ea$ of SRP uptake (eV) | 0.65 | 0.19 | 0.52 | 0.52 |
| Nutrient effect on respiration | none | none | none | 1.32× |





Table 2: Parameter estimates and model fit from laboratory experiments. In the first experiment, *Rhododendron maximum*
leaves were incubated at five temperatures ranging from 4-20ºC and rates of soluble reactive phosphorus (SRP) uptake and
respiration were measured. In the second experiment, *Acer rubrum* leaves were incubated with different initial concentrations
of phosphorus at different temperatures. We report slopes of the models we evaluated, the model $R^2$, and the *F*-value and *p*-
value associated with the slope parameter.

| Model | *Slope Estimate (SE)* | $R^2$ | *F* | *p* |
|---|---|---|---|---|
| *Experiment 1* | | | | |
| Respiration vs. temperature | $E_a$=1.02 (0.06) eV | 0.92 | $F_{1,24}$=292 | <0.0001 |
| SRP uptake vs. temperature | $E_a$=0.48 (0.05) eV | 0.81 | $F_{1,23}$=102 | <0.0001 |
| C:P vs. temperature | $E_a$=0.54 (0.06) eV | 0.69 | $F_{1,23}$=39 | 0.0002 |
| *Experiment 2* | | | | |
| Uptake $E_a$ vs. concentration | $K_s$ = 61.4 (48.6) µg L$^{-1}$, $U_{max}$= 0.56 (0.17) eV | 0.75 | NA | NA |
| $K_m$ vs. temperature | $E_a$=0.24 (0.24) eV | 0.00 | $F_{1,4}$=1.02 | 0.37 |
| $U_{max}$ vs. temperature | $E_a$=0.55 (0.15) eV | 0.69 | $F_{1,4}$=12.19 | 0.025 |

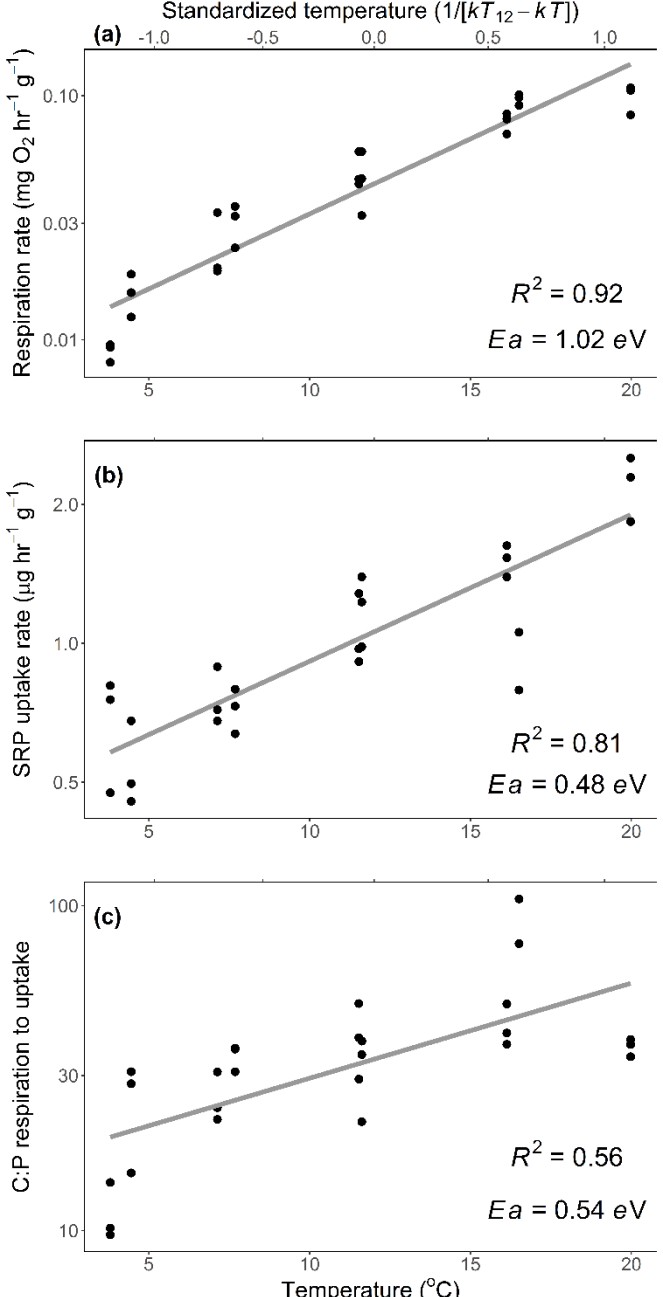


**Figure 1:** Mass-specific respiration (**a**) and soluble reactive phosphorus (SRP) uptake rates (**b**) of *Rhododendron maximum* leaves with temperature and (**c**) the molar ratio of C respired to P uptake across different temperatures. Standardized Boltzmann temperature is presented on the secondary *x*-axis. Points represent measurements from replicate bottles and grey lines represent best fits. Slopes of lines represent activation energies ($E_a$), which are reported in units of eV. Note *y*-axes are $\log_{10}$-scaled. See

Table 2 (experiment 1) for information on model fit, slopes, and significance.

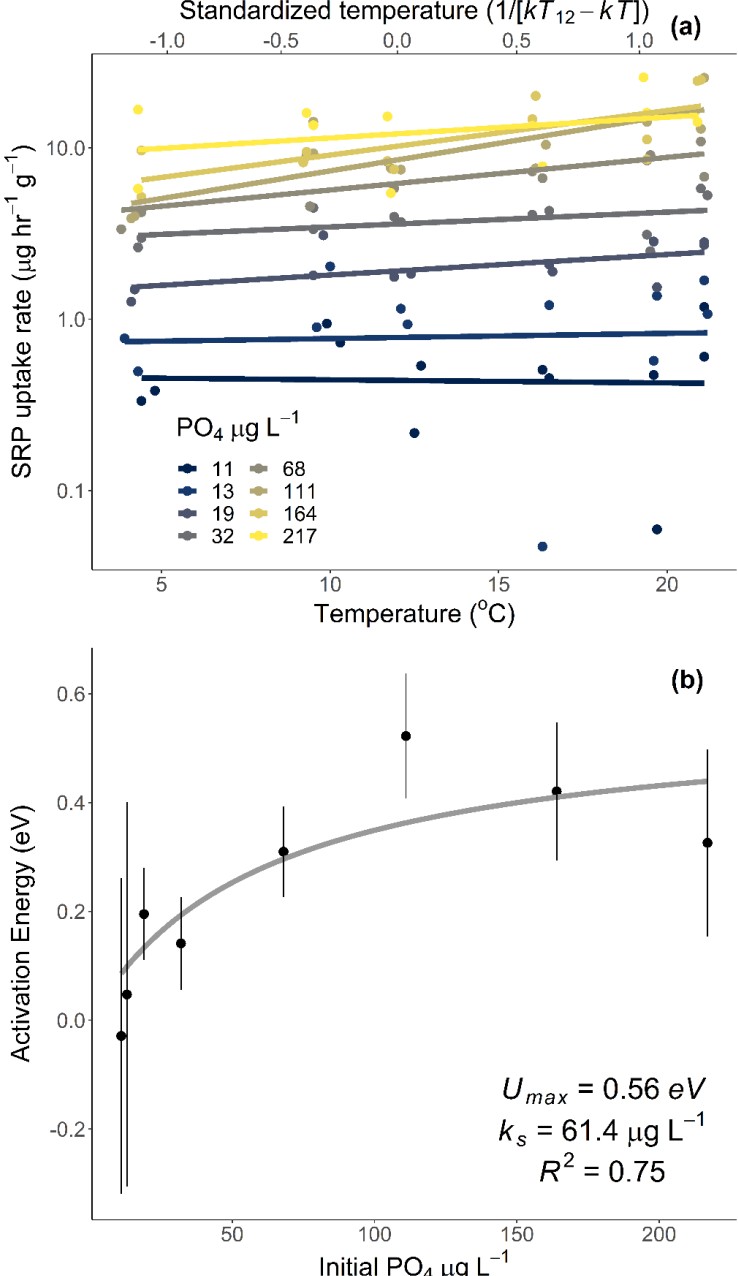

**Figure 2**: Rates of soluble reactive phosphorus (SRP) uptake compared to temperature for *Acer rubrum* leaves incubated at different temperatures and initial concentrations of SRP (**a**). The secondary *x*-axis in (**a**) represents the standardized Boltzmann temperature, and the *y*-axis is log₁₀-scaled. Slopes of the lines in (**a**) represent the activation energy of SRP uptake at different SRP concentrations. Equations for lines of best fit in (**a**) are given in Table A1. The slope estimates and their standard errors are plotted in (**b**). See Table 2 (experiment 2) for information on model fit and significance for (**b**).

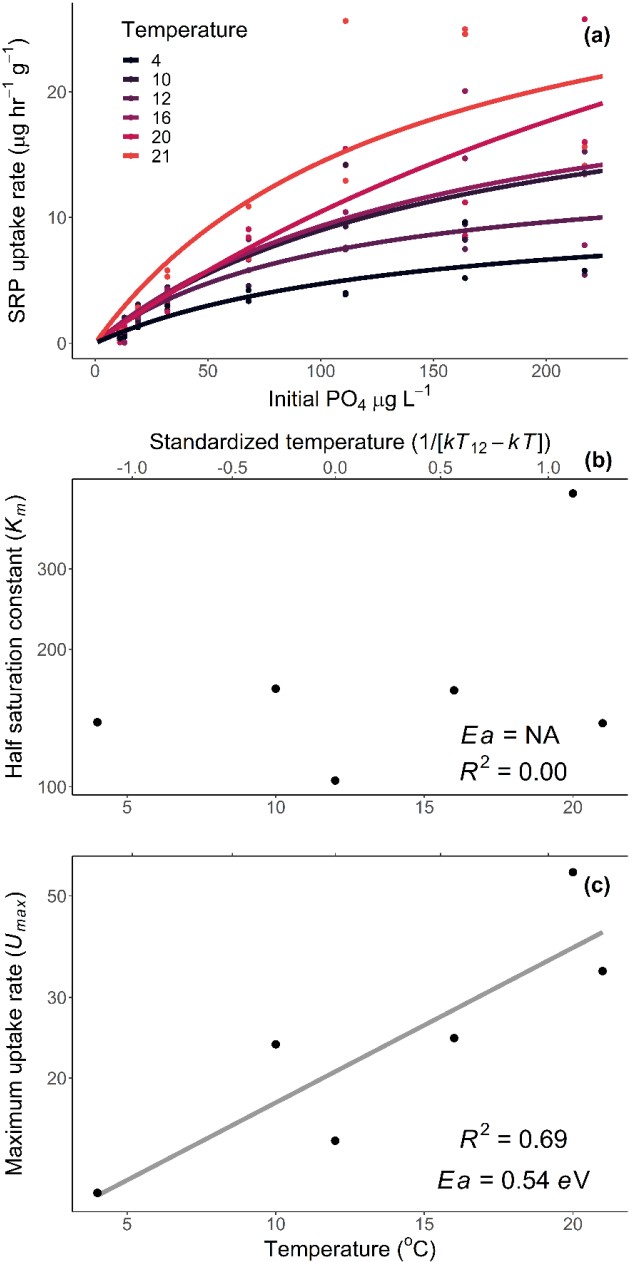

**Figure 3:** Uptake rates of soluble reactive phosphorus (SRP) across different initial concentrations of SRP grouped by
temperature (**a**). The lines represent the best-fit Michaelis-Menten kinetics, and the effect of temperature on the Michaelis-
Menten parameters, the half-saturation constant ($k_m$, **b**), and the maximum uptake rate ($U_{max}$, **c**) are represented in centered
Boltzmann-Arrhenius plots. In (**c**) the grey line indicates the best fit, which represents the activation energy in units of eV.

Equations and model fits for lines in (**a**) are given in Table A2. See Table 2 (experiment 2) for information on model fit and significance associated with (**b**) and (**c**).

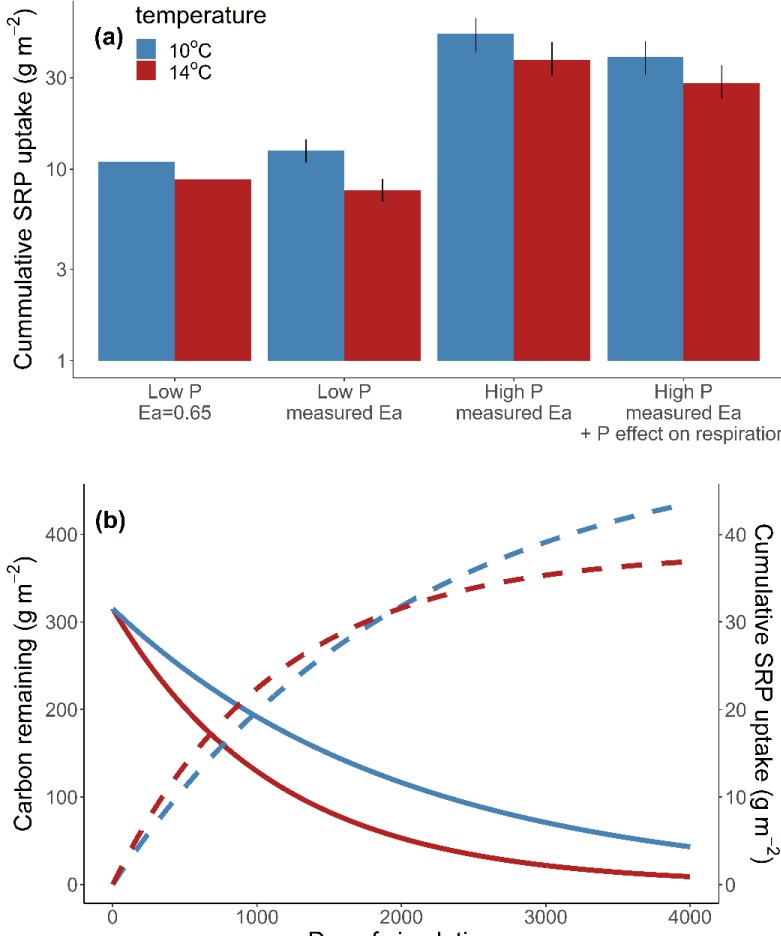

**Figure 4**: Simulations of the effect of temperature on cumulative soluble reactive phosphorus (SRP) uptake in four scenarios (**a**). First, we consider the effect of warming when the activation energy of respiration and uptake are both 0.65 eV (Low P, 0.65 eV). Second, we consider the effect of warming using the measured temperature dependence respiration and uptake at 19 µg L$^{-1}$ (Low P, measured $E_a$). Third, we consider the effect of warming at a high initial concentration of 111 µg L$^{-1}$ (High P, measured $E_a$). Finally, we considered the effect of warming at a high concentration where nutrients also affected the rates of respiration (High P, measured $E_a$ + P effect on respiration). Note the $y$-axis in (**a**) is log$_{10}$-transformed. We also include an example simulation from a cold and warm scenario using the temperature dependence from the high-SRP scenario (**b**). Mass of carbon (solid lines), and cumulative uptake of SRP (dashed lines), are presented over time for both temperatures, which are indicated by color. Model parameters used in each scenario are summarized in Table 1.

Table A1: Regression coefficients and model fits associated with Figure 2a in the main text. Slopes and intercepts are given along with the standard error (SE) of the estimates, the $p$-value associated with the slope, and the coefficient of variation ($R^2$).

| PO$_4$-P concentration ($\mu$g L$^{-1}$) | Intercept (SE) | Slope (SE) | $p$-value | $R^2$ |
|---|---|---|---|---|
| 11 | -0.82 (0.25) | -0.03 (0.29) | 0.92 | 0.00 |
| 13 | -0.25 (0.31) | 0.05 (0.35) | 0.90 | 0.00 |
| 19 | 0.65 (0.07) | 0.19 (0.08) | 0.04 | 0.35 |
| 32 | 1.28 (0.07) | 0.14 (0.09) | 0.13 | 0.21 |
| 68 | 1.83 (0.07) | 0.31 (0.08) | 0.00 | 0.58 |
| 111 | 2.15 (0.10) | 0.52 (0.11) | 0.00 | 0.67 |
| 164 | 2.33 (0.11) | 0.42 (0.13) | 0.01 | 0.52 |
| 217 | 2.40 (0.14) | 0.33 (0.17) | 0.09 | 0.29 |





Table A2: Regression coefficients and model fits associated with Figure 3a in the main text. Data were grouped by temperature and Michaelis-Menten models were fit to the data. Estimates of $K_m$ and $U_{max}$ along with the associated standard errors (SE) and $p$-values are presented here. We also present a pseudo-$R^2$ which is calculated as one minus the ratio of the residual sum of squares divided by the total sum of squares.

| Temperature | $k_m$ (SE) | $k_m$ ($p$-value) | $U_{max}$ (SE) | $U_{max}$ ($p$-value) | Pseudo-$R^2$ |
|---|---|---|---|---|---|
| 4 | 138.43 (89.10) | 0.14 | 11.23 (3.88) | 0.01 | 0.79 |
| 10 | 163.93 (91.11) | 0.09 | 23.69 (7.18) | 0.01 | 0.85 |
| 12 | 103.17 (64.06) | 0.13 | 14.59 (4.18) | 0.00 | 0.75 |
| 16 | 162.62 (121.89) | 0.20 | 24.46 (9.97) | 0.03 | 0.77 |
| 20 | 438.83 (510.76) | 0.40 | 56.31 (48.10) | 0.26 | 0.81 |
| 21 | 137.76 (101.98) | 0.20 | 34.24 (13.00) | 0.02 | 0.75 |

## Appendix 1: Duration of simulations

Objective:

In our simulations leaves take over 4000 days for 99% of the initial mass to decompose, and the cumulative uptake in the cold simulation exceeds the uptake in the warm simulation after ~1800 days into the simulation. The residence time of leaves in our simulations is much longer than has been observed in the field, where rhododendron leaves generally have a residence time of about two years (Manning *et al.* 2015). In this appendix we evaluate cumulative uptake in the warm and cold simulations when simulated respiration rates are increased to emulate more realistic rates of leaf breakdown.

Methods:

We simulated the breakdown of leaves and uptake of nutrients as described in the main text. The only change that was made to the simulation was altering the $r_{ref}$ of respiration from 0.04 mg $O_2$ g AFDM$^{-1}$ hr$^{-1}$ to 0.54 mg $O_2$ g AFDM$^{-1}$ hr$^{-1}$, which changes the rate of respiration at the reference temperature of 12°C. By changing this parameter we increase the rate that leaf mass is lost from the stream, without modifying how rates respond to changes in temperature. We simulated uptake and respiration using temperature sensitivities from the high soluble reactive phosphorus scenario, making the results of this simulation analogous to Figure 4b in the main text.

Results:

We found that in the simulations where leaf breakdown occurred at a more realistic rate there was still more cumulative uptake in the cold simulation than in the warm simulation. Uptake in the cold simulation was greater than uptake in the warm simulation after 139 days (Figure A1).

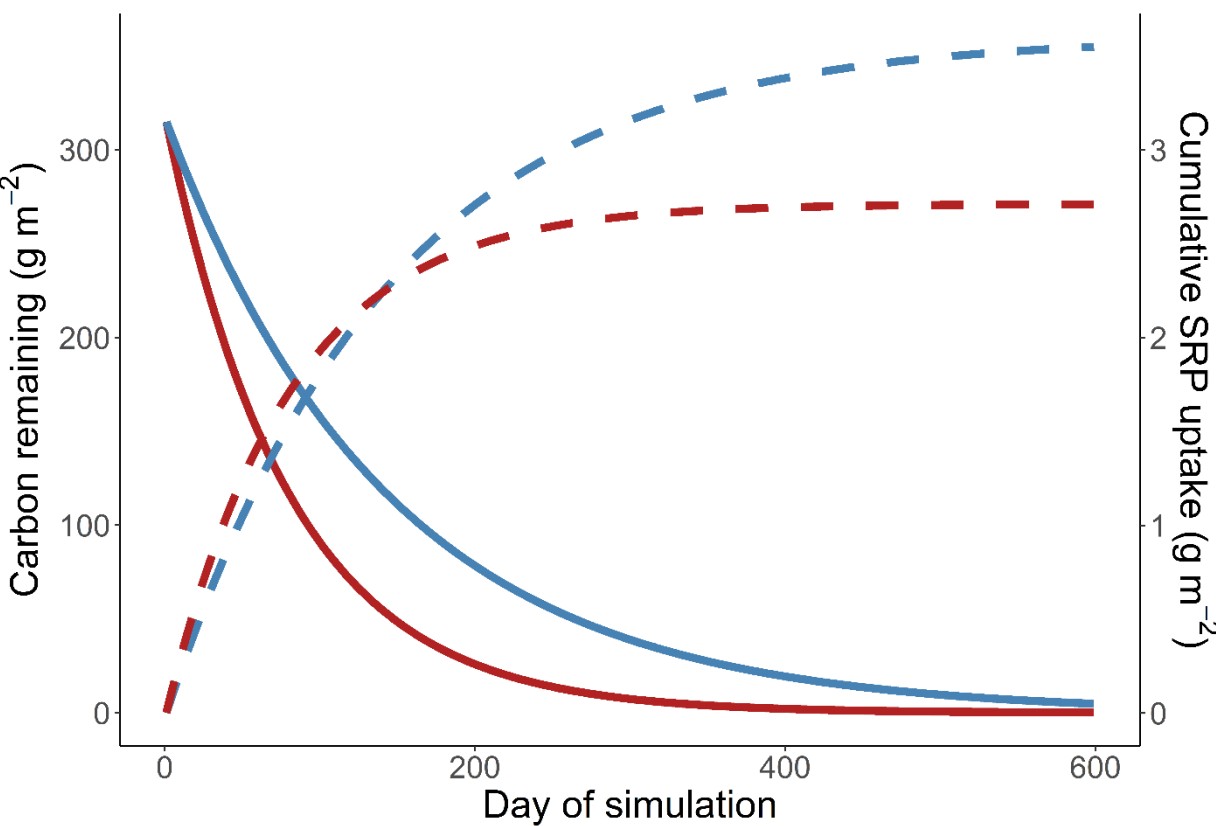

Figure A1: Simulation from a cold and warm scenario using a temperature dependence of respiration of 1.02 *eV*, a temperature dependence of soluble reactive phosphorus uptake of 0.52 *eV*, and a reference rate of respiration increased to make the residence times of carbon more realistic. Mass of carbon (solid lines), and cumulative uptake of SRP (dashed lines) are presented over time for a 14°C (red) and 10°C (blue) temperature scenario.



Appendix B2: Seasonal leaf inputs

Objective:

Our simulations do not incorporate the seasonal inputs of leaves that are an important feature of temperate forest ecosystems with deciduous vegetation. Here, we evaluate whether cumulative uptake in the cold scenario still exceeds cumulative uptake in a warm scenario when we incorporate seasonal inputs of leaf litter.

Methods:

We simulated the breakdown of leaves and uptake of nutrients as described in the main text. Once every 365 d an input of 315 g C m$^{-2}$ of leaves was added to the stream, which is the same amount that was initially supplied to the streams in our original simulation. We used the same simulation parameters as Appendix A, as they provide a better qualitative description of natural rates of leaf breakdown in the streams we are interested in representing here.

Results:

We found that cumulative uptake was greater in the cold scenario when we incorporated annual inputs of leaves into our simulations (Figure B1).

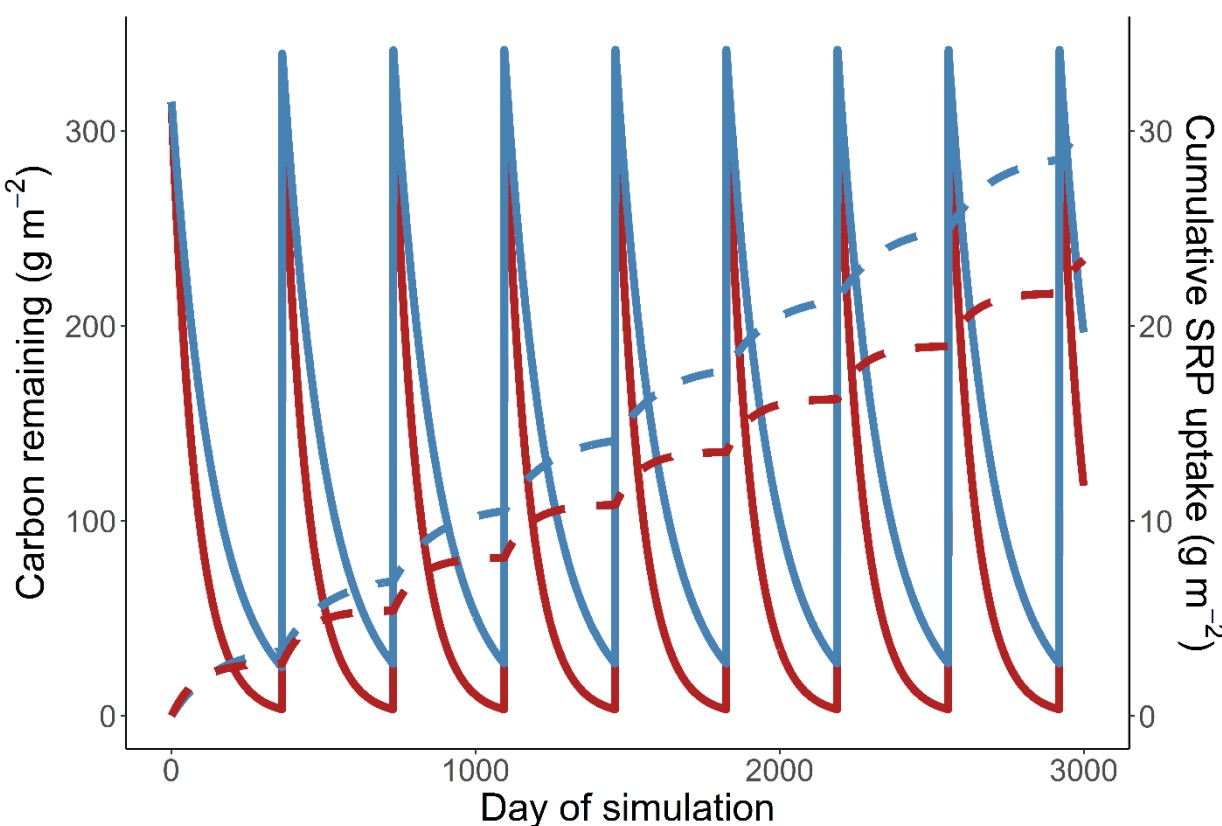

Figure B1: Simulation from a cold and warm scenario using a temperature dependence of respiration of 1.02 *eV*, a temperature dependence of soluble reactive phosphorus uptake of 0.52 *eV*, and a reference rate of respiration increased to make the residence times of carbon more realistic. These simulations also have annual leaf inputs which occur instantaneously, once every 365 days, and are equivalent to the initial mass of leaves in the stream (315 g C m$^{-2}$). Mass of carbon (solid lines), and cumulative uptake of SRP (dashed lines), are presented over time for a 14°C (red) and 10°C (blue) temperature scenario.
