# Peer review of "Contrasting activation energies of litter-associated respiration and P uptake drive lower cumulative P uptake at higher temperatures"

_Biogeosciences, 2022_

## Author Comment (AC1)

My main concern for discussion is the simulation exercise, which I think was a valuable addition but could be more realistic or its implications less overstated, particularly in the Abstract (L18-22). The stated aim of this simulation is to investigate the effects of C depletion on SRP uptake over the residence time of a pulsed leaf input. The residence time of this leaf is stated in the paper to be 2 years, but the simulation exercise extends for 3000 days. I note that in the example shown in Fig 4b, there is point at around 2000 days at which the cumulative SRP uptake in the colder treatment overtakes that of the warmer treatment. Prior to that, the implications of the simulation are opposite of the eventual conclusion: cumulative SRP uptake is higher in the warmer treatment (as C has not become depleted). Would ending the simulation at either 1 year (when another pulse of litter would become available in the following autumn) or 2 years (the typical residence time of this leaf) be a more appropriate time span to consider ecosystem-scale effects? Although the limitations of the simulation are well considered in the Discussion section, without building in the seasonal variation in litter availability and temperature the simulation may be too simple to inform the conclusions drawn in the Abstract.

There are three important points here that we want to address individually: 1.) how these results are framed in the abstract and title of the manuscript; 2.) the duration of the simulation and when to end it;, and 3.) how the seasonality of leaf litter additions may affect our overall conclusions drawn from the simulation modeling. We address these in turn below.

1. *Framing of simulations*. When drafting this manuscript, we were seeking concise language to differentiate between the instantaneous measurements we made in the lab, and the long-term cumulative uptake dynamics that we simulate. In part because we express the simulation models on an areal basis, we had settled on using "ecosystem" as a descriptor of these rates. However, this comment illustrates that different language would more clearly communicate the results and implications of our study. In revising this manuscript, we are proposing to change the phrase "ecosystem uptake" to "long-term cumulative uptake," as this describes our results more specifically. Most importantly, we don't intend to imply that our modeled results are capturing the full complexity of nutrient uptake at the ecosystem scale.

2. *The duration of the simulation models*. The reviewer is correct that if we chose to end our simulation models around the one- or two-year mark then our conclusions from the simulations would be opposite. However, the slow breakdown in our models indicates that the leaves are roughly one-third to one-half broken down at those points in time – so they are not very close to their residence time in our simulations. We discuss some possible reasons for the long residence times in the paper (Page 9, lines 271-276), and don't contend that our modeled rates represent the actual residence time of these leaves in a stream. If we reduce the residence time of leaves in our models to more closely match observations in the field (by increasing the base rate of respiration, not the response of respiration to temperature), then we see the lines representing cumulative uptake cross at around 100 days into the simulation (instead of at day 2000). We would be happy to include some of these "ambient speed" versions of the simulations in an appendix of a future version of this manuscript and elaborate on this further in our discussion.

3. *Seasonality*. In temperate climates with deciduous vegetation, leaf inputs to streams are highly concentrated at the end of the growing season. Thus, after about a year there would be more leaves added that are also taking up nutrients and contributing to

ecosystem nutrient uptake. This would mean that uptake rates in both the high and lower temperature simulation would increase. Our intention, and how we would like to refocus the paper in a revision, is on the behavior of a cohort of leaves over its residence time in a stream. We hope that the above-mentioned changes to the wording used to describe the results of our simulations will help make the intended scope of our modeling clearer. Including seasonality would be important for predicting ecosystem-level rates through time – but we don't believe it is necessary to understand the overall long-term effects of temperature on cumulative uptake rates. Our models indicate that in the warmer simulation each gram of leaf is taking up less nutrients over its residence time, and this wouldn't be changed by adding seasonal inputs (although areal rates would indeed change). Over a long enough simulation, the inputs of leaves would reach a dynamic equilibrium with decomposition. Because in the warm simulation each gram of leaves is taking up less nutrients over its residence time, the general finding of lower cumulative uptake in the warmer simulation should hold with the addition of seasonal inputs to the model. We would be happy to include seasonal simulations in an appendix of a future version of this manuscript to confirm that our main conclusion is robust to seasonal leaf inputs.

**Specific comments:**

Line 34: I don't quite follow the logic here of the comparison to an autotrophic system. Does this line refer to an increase is autotroph growth or heterotrophic microbial growth? An increase in growth/biomass in any case would lead to a higher demand for nutrients.

Our intention was draw a distinction between donor-controlled systems, like forest streams, and systems in which the primary energy inputs are from *in situ* primary production. In revising this manuscript we would like to clarify this by changing the line to read "In autotrophic systems, increases in temperature drive increases in primary production that result in predictably higher demand for nutrients (Rasmussen et al., 2011); however, in donor-controlled detrital systems, such as soils and forest streams, increased rates of metabolism stimulated by increases in temperature or nutrients can lead to reductions in pools of the dead organic matter that fuels metabolism, eventually reducing microbial biomass (Walker et al., 2018; Suberkropp et al., 2010)."

Line 70: Might be worth clarifying this is the case for temperate systems in the northern hemisphere that have deciduous riparian vegetation.

Thanks for pointing this out. In revising our manuscript, we would like to acknowledge that this mainly applies to deciduous vegetation. We would like to change this line to read "In temperate ecosystems with deciduous vegetation, there is strong seasonality in the input of senescent leaf litter inputs."

L179: Is the 250 mg C m$^{-2}$ based on observations from the catchment or a similar one? Are there measurements to provide a typical mass of detrital leaf litter in each season?

250 mg C m$^{-2}$ is a little lower than is typically observed as the sites where we incubated leaves for this study, but the total mass of leaves at the start of the simulation should not alter the main findings of the simulation models. In revising this manuscript, we plan to use values of initial input of leaves from Suberkropp et al. (2010), which also provides monthly estimates of leaf litter standing stocks for more than five years.

L190: Could the parameters of these different scenarios perhaps be presented in a table? It is difficult to compare from this text (although it is clearer in Fig 4a).

We would be happy to add this table to a revised version of the manuscript.

Figure 4b: The methods indicated the simulation started at 250 mg C m$^{-2}$, however the y axis here begins at 150 mg C m$^{-2}$ at day 0. Please clarify.

Thank you for catching this, which was a mistake in the analysis. We had made a conversion from grams of ash-free dry mass to g C for the initial mass of leaves in the stream, which we did not need to do. We plan to fix this mistake in a revision of the manuscript, and it will not impact the major findings from the simulations.

---

## Author Comment (AC2)

While I think this was a well conceived, designed, and conducted study, I believe that the authors may have pushed their results a bit farther than warranted. Particularly given the scope of the laboratory study. The simulation results were used to compare to reach-scale studies despite the simulation ignoring a multitude of other P uptake and respiration mechanisms common in streams and likely affected by temperature in complex ways. I don't think the simulation study is a problem, but I think it should be discussed for what it is: a simple simulation to develop testable hypotheses. It is not representative of real world expectations and shouldn't be considered as such. There were also some issues with the methods and particularly the lack of detail on statistical analyses performed. I think this manuscript has the potential to be a valuable contributor to the field and poses some interesting next steps that must be considered as we continue to push further into coupled C-N-P cycles and expectations with warmer climates.

We acknowledge that our simulation models only encompass a limited proportion of the total uptake that occurs in a forest stream and that we don't include abiotic sorption, uptake by primary producers, or uptake by microbes on fine particles, wood, or buried sediments. In revising this manuscript, we would like to modify the language we use to describe the results of our simulations to more specifically describe what they represent. We would like to frame the results of our simulations as describing "long-term cumulative uptake" and not "Ecosystem uptake". Additionally, in revising the manuscript we plan to be more careful in how we state the implications of the findings from our experiments and simulations.

Specific comments:

Line 16: I don't know what a 0.48 and 1.02 eV value means for temperature dependence. Is this standard unit/metric used to compare temperature dependence of various processes? Not sure if this is the best choice for the abstract.

Quantifying the effect of temperature on rates using the activation energy from the Boltzmann-Arrhenius equation is the convention in the field of metabolic ecology (e.g., Brown et al 2004, and other citations in the manuscript), and eV is a standard unit of kinetic energy.

Line 17: for ranges (0.12 to 0.48, or 11 to 212) I encourage authors to use "to" instead of a hyphen because a hyphen could be misconstrued to represent a negative sign.

We can make this change when revising the manuscript.

Line 34: Should this be increases in productivity rather than increases in growth? The Rasmussen citation quantified stream metabolic activity (GPP, ER) not growth.

The reviewer is correct. This should be increases in productivity.

Line 40: If the authors are using U in the nutrient spiraling sense, isn't U directly correlated with nutrient availability? It's in the calculation, isn't it?

We agree that there is good information about the effect of concentration on rates of nutrient uptake. In revising this manuscript, we would like to revise this topic sentence to clarify that we

mean that the joint effects of temperature and nutrient concentration on uptake are unknown. We plan to revise this sentence to read: "Mechanisms explaining the joint effects of temperature and nutrients on mass-specific rates of nutrient uptake ($U$) remain poorly resolved."

Lines 60 – 61: I mean, maybe? But the Michaelis-Menten kinetics (Vmax, Ks, etc) might not kick in until super high concentrations, though.

We plan to make this statement less definitive by revising this passage to read "Consequently, the proportion of dissolved nutrients taken up by the microbial community may decline with increasing nutrient concentration (O'Brien et al., 2007)."

Lines 76-78: How much of this negative effect of temperature could be due to canopies opening up in cooler months leading to more light and subsequently more autotrophic nutrient uptake? Even forested headwater streams have open canopies sometime and that short window of autotrophic activity could offset heterotrophic decreases maybe?

We thank the reviewer for pointing this out. This same point comes up again later in comments on the discussion. We address this comment there.

Lines 103 – 104: When were bags deployed initially? There are two collection dates but only one incubation date. Not a big deal but if you are going to report collection date, report deployment date, too.

The bags were deployed on 17 November 2020. We can add this date to a revised version of the manuscript.

Line 105: Were fragments a consistent size? Or was there a targeted size? Why cut the leaves into smaller fragments? How much leaf litter material (mass) was added to each bottle? How much water (1L?)

The leaf fragments were cut to a target size of 1.5 cm x 1.5 cm squares. The leaves were cut to allow measurements in smaller volumes of water. We do not have an estimate of the mass of leaves in each 1-L bottle. We added approximately 1 L of water to each bottle. We can add this information to the methods in a revised version of this manuscript.

Line 113: Were blanks measured initially and after the end of the incubation the same way? I worry about displacement/replacement of water due to the initial DO measurement given how large DO probes can be compared to a scint vial.

The blanks and samples were measured the same way before and after the experiment. Roughly 15% of the water in the vial was displaced during the measurement of the initial DO concentration, and it was immediately replaced with streamwater from the same source that was used to fill the vial. We can add this information to the methods when revising the manuscript.

Line 113: Was there at least an attempt to add a similar amount of leaf material to each vial?

Yes, we attempted to add a similar amount of leaf material to each vial.

Line 117: I suggest writing this out as an equation. Were the incubations done in the dark? I don't see anywhere suggesting that. If incubations were not done in the dark then this approach yields NEP, not respiration.

We can add an equation to a revised version of the manuscript. The incubations were done in the dark. We will add this information to a revised version of the manuscript.

Line 123: nominal pore size?

The nominal pore size was 1.0 micron. We can include this information in a revised version of the manuscript.

Line 126: U is traditionally reported in units of mass per area per time (e.g., mg P / m2 / h).  I think the approach the authors have taken here to estimate uptake as mass of nutrient per mass of leaf per time is fine but I think that something other than just "U" should be used here. Also, I'm not the biggest fan of the calculation for U as I would greatly prefer an initial and final sample collected from the same sample container. The authors are assuming that all incubation vials started with the same conditions. I don't know how I feel about that assumption. Were individual tubes amended with P? Or was a reservoir amended with P and then added to the tubes? Also, the drastic differences in incubation time is strongly suggesting an assumption of linear P uptake which I don't know can be expected to hold true across different concentrations.

For the sake of clarity, we would like to use U, which we define clearly at first usage. For each temperature treatment a reservoir of streamwater was amended with P, shaken to mix the solution, and then this nutrient-amended water was added to the vials that were incubated. Our analysis does assume that all vials within a given temperature treatment had the same initial concentration. The three blanks that were measured from each temperature treatment generally had measurement of P concentration that were within 1-2 $\mu$g L$^{-1}$ of one another, which gives us confidence that the reservoir was well mixed, and that the initial conditions were similar among vials. The approach we used allowed us to forgo removing water to measure P concentrations at the beginning of the experiment. When designing this experiment, we were concerned that taking an initial sample from each vial would change water volumes and could have a large effect on rates given the small volumes of water we used. We used shorter incubation times for the warmer incubations in this experiment because we expected uptake and respiration to proceed more quickly. Our hope was that similar masses of P would be taken up across the different temperature treatments to limit the importance of kinetic effects.

Line 135: Where was this categorical block effect included? What are the different experimental batches? I don't understand this statement at all.

We ran the experiment twice to increase our sample size and so wanted to account for any differences between the first and second batch of the experiment in our statistical models. There may have been differences in the microbial communities, the streamwater we collected, or other factors between the two batches of the experiment we ran, and we wanted to account for these in

our statistical models. The block effect was included in the statistical model by adding a categorical variable that indicated which batch each measurement was from. In practice, this meant that the model estimated a different *y*-intercept for the data from each batch, and a single slope. We plan to modify this sentence to make our statistical design clearer: "The two dates the experiment was run on may have had differing biological or environmental conditions, so we included a categorical effect of date in our statistical models to account for any differences."

Lines 136 – 138: What? I do not understand this statement at all. There was a model that compared respiration and Usrp to each other and a categorical variable that indicated if the model was for respiration or Usrp? What kind of a model? How was it evaluated? What was the dependent variable? The dependent variables? More detail and description is needed here.

The model compared the slopes of the relationship between temperature and each rate (respiration and uptake), which are the activation energies. The model is essentially an ANCOVA, but with temperature transformed so that slopes represent activation energies. Because the question we are testing is whether the slopes are different, we are interested in evaluating the significance of the interaction term in this model. We plan to modify this sentence to: "We used an ANCOVA-type linear model to evaluate whether the response of respiration and Usrp to temperature were different. We fit a linear model that described the natural log-transformed rates of respiration or uptake rates as a function of the standardized Boltzmann temperature. The model included an interaction between temperature and the type of rate (i.e., respiration or uptake). A significant interaction term in this model indicates that the slopes of the relationships between temperature and these rates differ."

Line 138: A significant interaction term between what?

The interaction term in the model is between temperature and a categorical variable that describes whether the rate is respiration or uptake (see revised text above).

Line 143: Why was the centered inverse Boltzmann temperature used as the predictor variable? Why not just temperature?

By regressing natural log-transformed rates of biological processes against the inverse Boltzmann temperature the slopes of these relationships can be interpreted biologically as an activation energy. We plan to add more description of this to a revised version of the manuscript.

Line 149: Were leaves weighed at the end of the experiment tin the same manner? How were initial SRP concentrations achieved?

These methods were the same as those we described above for the first experiment. The leaves were weighed and the P concentrations were achieved by adding a concentrated solution of phosphorus to streamwater. We will add this detail to a revised version of the text.

Lines 150 – 154: So 6 temperatures * 8 SRP concentrations * 3 bottles per treatment = 144 individual incubations. Is that accurate?

Yes, that is accurate.

Lines 145 – 173: So basically Usrp was regressed against temperature for each initial SRP concentration and then the slope of those regressions were compared across initial SRP concentration? Was a regression or correlation or something done here? There don't seem to be any stats, it reads like the authors plotted these out and visualized them but that's not a real satisfying analysis in my opinion. The same general though holds for the M-M analysis, too.

The regressions that we did in the section were not just visual - we fit regression models to the data. We can include the parameters and model fits in an appendix of a revised version of the manuscript.

Line 199: It seems like there should to be an analysis section. Or more detail needs to be given for the analyses in the individual sections (as was described in some of my previous comments). How were the simulation models assessed/evaluated?

Because there are three distinct units (two experiments and the simulations), we prefer to include the relevant analysis within each section, instead of describing all the analyses in one unified section. We evaluated the outcomes of the simulations by simply comparing the estimated effects. We did not include statistics for this section. We can clarify this in the revised text of the manuscript.

Line 203: canonical is an odd word choice here.

We can drop this word from a revised version of the manuscript, although this word is often used in the metabolic theory literature to describe this important value of the activation energy of cellular respiration.

Line 226: These are interesting results. It's definitely a very simplified model, but I think that is acknowledged and it points towards interesting (and testable) mechanisms changing P dynamics with future warming expectations. Obviously there are many other things to consider (e.g., changes in animal behaviors altering the decomposition of leaves, shifts in phenology matching shifts in climate, changes in N dynamics and broader stoichiometric questions…) but still an interesting exercise.

Thank you, we agree that our approach is simplified in terms of parts of the ecosystem that we model. We appreciate that other studies could add additional components such as animal behavior, phenology, and broader stoichiometric issues.

Line 256: Again, I wonder how much temperature is correlated with canopy cover in some of these whole system nutrient spiraling studies. I also think it's difficult to compare a scint vial's worth of Usrp to a full stream nutrient release. The authors have quantified the effect of temperature on leaf respiration and leaf-based Usrp. They did not measure anything about other components of the ecosystem that could/would change with temperature (e.g., sediment uptake dynamics, hyporheic processes, autotrophic uptake (which would increase with decreasing temps due to autocorrelation with canopy cover). While I think the simulation study was a valuable

exercise, I don't think that these results can really be extrapolated and compared to reach-scale results/studies.

We plan to add acknowledgement of the potential role for autotrophic nutrient uptake (and other processes) in modifying the trends we observed to this section of the manuscript. While we don't contend that the rates we measure are equivalent to rates measured at the whole-stream scale, we do think it is useful to compare the uptake measurements we made in the laboratory with uptake measurements made at the stream-reach scale.

Lines 262 – 263: This statement is unfounded. I disagree that the current study separated the contribution of physiological and biomass-mediated effects of temperature on ecosystem-level nutrient uptake. As mentioned in the previous comment, the study separated the contribution of these temperature-based mechanisms to affect leaf litter respiration and Usrp. Even in the most detritally-driven ecosystems (of which, Coweeta stream are definitely up near the top), there are still a multitude of other autotrohic and heterotrophic compartments contributing to ecosystem-scale respiration and nutrient uptake. This section should be either deleted or modified to be more accurate for what the study actually did do (which is still a valiant effort!).

We appreciate this point and would like to change how we refer to the results of the simulation throughout the paper. As we mentioned above, we would like to change "ecosystem-level nutrient uptake" to "long-term cumulative uptake". In that narrower context we believe that we do separate these direct and indirect effects.

Line 265: But this ignores potential increases in sediment-based Usrp. Or autotrophic. Or hyporheric. Maybe the leaf litter breakdown is fueling more labile DOM to reach interstitial spaces where sorbed P can be broken down.

We agree that our model does not include these factors. In revising this manuscript, we plan to define the results of our simulations more narrowly, which should address these concerns.

Line 295: These are great caveats to include. I don't know how the authors can make the bold claims such as in lines 262 – 263 and then simultaneously acknowledge all of these issues.

In revising this manuscript, we would like to describe the results of our simulations using language that is more specific, as we have described above.

Line 305:Not sure how this study addresses/provides insight into the effect of observational scale on temperature sensitivity.

The comparison we are looking at drawing here is between the measured instantaneous effects and the modeled long-term effects. In revising this manuscript, we will change this from "observational scale" to "time scale".

Line 305: The majority (entirety?) of the discussion focuses on the simulation experiment, which is the weakest part of this paper in my opinion. I think more general discussion of temperature dependence of biogeochemical processes and how that can affect things more broadly would

worth including initially, and then a toned down version of the focus on the simulation model. E.g. 'The results of our lab incubations would theoretically imply xyz. Our simulation studies confirm some of these expectations but revealed somewhat contradictory patterns due to abc.'

We note that this reviewer has the perception that the discussion focused too much on the simulation model in this version of the manuscript. We will try to seek a better balance when revising the manuscript.

Figure 1: I know the stats and fits are included in table 1, but I still think they'd be good to include on the figures, personally. I recommend including the equations for each line as well as stats (r-2, p-value).

When revising the manuscript, we can add this information to the figure caption.

Figure 2a: Are each of these significant regressions in panel a? Doesn't seem like it, which would entail no relationship between SRP uptake and temperature at certain initial values. A table (supplemental?) with slopes, r-squared, etc supporting these model fits would be useful.  I'm not sure what in table 1 is showing model fit for these lines. Maybe a more details statistical analysis section in the methods would help clarify things a bit.

In a revised version of the manuscript, we can add a table of the model fits in equation 2a and 3a to the appendix, and revise the figure captions.

Figure 2b: It almost looks like this is a hump-shaped relationship maxing out at mid-concentration. Inhibitory effect of high P? Any reason this particular form of curve was used?

If there was an inhibitory effect of high P, then we would expect lower rates of P uptake at all temperatures in the highest P concentration. We don't really see this in Figure 2a, where we visualize the rates directly. In Figure 2b we see that the response of P uptake to temperature does peak at intermediate concentrations, but we don't think that inhibition at high P can explain this. We used this fit because saturating functions are often used to describe the effect of increasing concentration on ecological processes.

Figure 3: Now by the third figure, I'm really having difficulty connecting individual panels from figures 1 – 3 to stats and model fits from table 1. Make it easy for me (and the other readers) by putting this info on the individual panels.

When revising the manuscript, we can add this information to the figure caption.

---

## Author Response (AR1)

Thank you for the opportunity to revise this manuscript. Please see our point by point response to reviewers below.

Reviewer 1:

My main concern for discussion is the simulation exercise, which I think was a valuable addition but could be more realistic or its implications less overstated, particularly in the Abstract (L18-22). The stated aim of this simulation is to investigate the effects of C depletion on SRP uptake over the residence time of a pulsed leaf input. The residence time of this leaf is stated in the paper to be 2 years, but the simulation exercise extends for 3000 days. I note that in the example shown in Fig 4b, there is point at around 2000 days at which the cumulative SRP uptake in the colder treatment overtakes that of the warmer treatment. Prior to that, the implications of the simulation are opposite of the eventual conclusion: cumulative SRP uptake is higher in the warmer treatment (as C has not become depleted). Would ending the simulation at either 1 year (when another pulse of litter would become available in the following autumn) or 2 years (the typical residence time of this leaf) be a more appropriate time span to consider ecosystem-scale effects? Although the limitations of the simulation are well considered in the Discussion section, without building in the seasonal variation in litter availability and temperature the simulation may be too simple to inform the conclusions drawn in the Abstract.

There are three important points here that we would like to address individually: 1.) how these results are framed in the abstract and title of the manuscript; 2.) the duration of the simulation and when it is most appropriate to end it;, and 3.) how the seasonality of leaf litter additions may affect our overall conclusions drawn from the simulation modeling. We address these in turn below.

1.  *Framing of simulations*. When drafting this manuscript, we were seeking concise language to differentiate between the instantaneous measurements we made in the lab, and the long-term cumulative uptake dynamics that we simulate. In part because we express the simulation models on an areal basis, we had settled on using "ecosystem" as a descriptor of these rates. However, this comment illustrates that different language would more clearly communicate the results and implications of our study. In revising this manuscript, we have changed from using the phrase "ecosystem uptake" to "long-term cumulative uptake," as this describes our results more specifically. Most importantly, in revising the manuscript we have been careful to not imply that our modeled results are capturing the full complexity of nutrient uptake at the ecosystem level.
2.  *The duration of the simulation models*. The reviewer is correct that if we chose to end our simulation models around the one- or two-year mark then our conclusions from the simulations would be opposite. However, the slow breakdown in our models indicates that the leaves are roughly one-third to one-half broken down at those points in time – so they are not very close to their residence time in our simulations. We discuss some possible reasons for the long residence times in the paper (Page 10, lines 286-296), and don't contend that our modeled rates represent the actual residence time of these leaves in a stream. If we reduce the residence time of leaves in our models to more closely match

observations in the field (by increasing the reference rate of respiration, not the response of respiration to temperature), then we see the lines representing cumulative uptake cross at 139 days into the simulation (instead of at day 2000). We have added simulations with more realistic rates of leaf breakdown to the supplemental material (Appendix 1). And reference these results in the discussion: "Increasing rates of breakdown in our simulations to mimic residence times observed in the field leads to cumulative uptake in the cool scenario exceeding cumulative uptake in the warm scenario after only 139 days (Appendix S1)" Page 10, Lines 294-296

3. *Seasonality*. In temperate climates with deciduous vegetation, leaf inputs to streams are highly concentrated at the end of the growing season. Thus, after about a year there would be more leaves added that are also taking up nutrients and contributing to ecosystem nutrient uptake. This would mean that uptake rates in both the warm and cool temperature simulation would increase. Our intention with these simulations, and how we would like to refocus our discussion of these results in the paper, is on the behavior of a cohort of leaves over its residence time in a stream. We hope that the above-mentioned changes to the wording used to describe the results of our simulations will help make the intended scope of our modeling clearer. Including seasonality would be important for predicting ecosystem-level rates through time – but we don't believe it is necessary to understand the overall long-term effects of temperature on cumulative uptake rates. Our models indicate that in the warmer simulation each gram of leaf is taking up less nutrients over its residence time, and this wouldn't be changed by adding seasonal inputs (although areal rates would indeed change). We have included simulations with seasonal inputs in Appendix S2 that demonstrate that cumulative uptake in the cool scenario still exceeds cumulative uptake in the warm scenario when seasonal inputs are incorporated. We mention these findings in the discussion: "Furthermore, while our models did not include seasonal inputs of leaves, our general finding that cumulative uptake in cool scenarios is greater than in warm scenarios is robust to successive seasonal inputs of leaves (Appendix S2)" Page 10, Lines 296-298

**Specific comments:**

Line 34: I don't quite follow the logic here of the comparison to an autotrophic system. Does this line refer to an increase is autotroph growth or heterotrophic microbial growth? An increase in growth/biomass in any case would lead to a higher demand for nutrients.

Our intention was to draw a distinction between donor-controlled systems, like forest streams, and systems in which the primary energy inputs are from *in situ* primary production. In revising this manuscript we would like to clarify this by changing the line to read "In autotrophic systems, increases in temperature drive increases in gross primary production, resulting in predictably higher demand for nutrients (Rasmussen et al., 2011); however, in donor-controlled detrital systems, such as soils and forest streams, increased rates of metabolism stimulated by increases in temperature or nutrients can lead to reductions in pools of the dead organic matter that fuels metabolism, eventually reducing microbial biomass on an areal basis (Walker et al., 2018; Suberkropp et al., 2010)." Page 2, Lines 33-37

Line 70: Might be worth clarifying this is the case for temperate systems in the northern hemisphere that have deciduous riparian vegetation.

Thanks for pointing this out. This line now reads: "In temperate ecosystems with deciduous vegetation, there is strong seasonality in the input of senescent leaf litter." Page 3, Lines 71-72

L179: Is the 250 mg C m$^{-2}$ based on observations from the catchment or a similar one? Are there measurements to provide a typical mass of detrital leaf litter in each season?

We have revised the mass of leaves used in the simulation to match values of peak leaf litter standing stocks observed at the Coweeta Hydrologic lab. This line now reads: "The simulated stream reach starts with 315 g leaf C m$^{-2}$, which is based on observations of leaf standing stocks in streams at CHL (Suberkropp et al., 2010)" Page 7, Lines 192-193

L190: Could the parameters of these different scenarios perhaps be presented in a table? It is difficult to compare from this text (although it is clearer in Fig 4a).

We have added this table to the manuscript as a new Table 1.

Figure 4b: The methods indicated the simulation started at 250 mg C m$^{-2}$, however the y axis here begins at 150 mg C m$^{-2}$ at day 0. Please clarify.

Thank you for catching this, which was a mistake in the analysis. We had made a conversion from grams of ash-free dry mass to g C for the initial mass of leaves in the stream, which we did not need to do. We have fixed this mistake in this revision, and it has not changed the major findings from the simulations.

Reviewer 2:

While I think this was a well conceived, designed, and conducted study, I believe that the authors may have pushed their results a bit farther than warranted. Particularly given the scope of the laboratory study. The simulation results were used to compare to reach-scale studies despite the simulation ignoring a multitude of other P uptake and respiration mechanisms common in streams and likely affected by temperature in complex ways. I don't think the simulation study is a problem, but I think it should be discussed for what it is: a simple simulation to develop testable hypotheses. It is not representative of real world expectations and shouldn't be considered as such. There were also some issues with the methods and particularly the lack of detail on statistical analyses performed. I think this manuscript has the potential to be a valuable contributor to the field and poses some interesting next steps that must be considered as we continue to push further into coupled C-N-P cycles and expectations with warmer climates.

We acknowledge that our simulation models only encompass a limited proportion of the total uptake that occurs in a forest stream and that we don't include abiotic sorption, uptake by primary producers, or uptake by microbes on fine particles, wood, or buried sediments. In revising this manuscript, have modified the language we use to describe the results of our

simulations to be more specific in describing what they represent. We have reframed the results of our simulations as describing "long-term cumulative uptake" and not "Ecosystem uptake".

Specific comments:

Line 16: I don't know what a 0.48 and 1.02 eV value means for temperature dependence. Is this standard unit/metric used to compare temperature dependence of various processes? Not sure if this is the best choice for the abstract.

Quantifying the effect of temperature on rates using the activation energy from the Boltzmann-Arrhenius equation is the convention in the field of metabolic ecology (e.g., Brown et al 2004, and other citations in the manuscript), and eV is a standard unit of kinetic energy.

Line 17: for ranges (0.12 to 0.48, or 11 to 212) I encourage authors to use "to" instead of a hyphen because a hyphen could be misconstrued to represent a negative sign.

We have made this change throughout when revising the manuscript.

Line 34: Should this be increases in productivity rather than increases in growth? The Rasmussen citation quantified stream metabolic activity (GPP, ER) not growth.

This is correct. This line now reads "In autotrophic systems, increases in temperature drive increases in gross primary production, resulting in predictably higher demand for nutrients (Rasmussen et al., 2011); however, in donor-controlled detrital systems, such as soils and forest streams, increased rates of metabolism stimulated by increases in temperature or nutrients can lead to reductions in pools of the dead organic matter that fuels metabolism, eventually reducing microbial biomass on an areal basis(Walker et al., 2018; Suberkropp et al., 2010)" Page 2 Lines 33-38

Line 40: If the authors are using U in the nutrient spiraling sense, isn't U directly correlated with nutrient availability? It's in the calculation, isn't it?

We agree that there is good information about the effect of concentration on rates of nutrient uptake. In revising this manuscript, we have clarified that we mean that the joint effects of temperature and nutrient concentration on uptake are unknown. The text now reads: "Mechanisms explaining the joint effects of temperature and nutrients on mass-specific rates of nutrient uptake ($U$) remain poorly resolved." Page 2, Line 41-42

Lines 60 – 61: I mean, maybe? But the Michaelis-Menten kinetics (Vmax, Ks, etc) might not kick in until super high concentrations, though.

We have revised this sentence to make our statement less definitive: "Consequently, the proportion of dissolved nutrients taken up by the microbial community may decline with increasing nutrient concentration (O'Brien et al., 2007)." Page 3, Lines 61-62

Lines 76-78: How much of this negative effect of temperature could be due to canopies opening up in cooler months leading to more light and subsequently more autotrophic nutrient uptake? Even forested headwater streams have open canopies sometime and that short window of autotrophic activity could offset heterotrophic decreases maybe?

We thank the reviewer for pointing this out. This same point comes up again later in comments on the discussion. We address this comment there.

Lines 103 – 104: When were bags deployed initially? There are two collection dates but only one incubation date. Not a big deal but if you are going to report collection date, report deployment date, too.

The bags were deployed on 17 November 2020. This line now reads: "We incubated *Rhododendron maximum* (hereafter, *Rhododendron*) leaf litter to allow for microbial colonization in Watershed 5a in 5-mm mesh litterbags for 114 days beginning on 17 November 2020" Page 4, Lines 103-105

Line 105: Were fragments a consistent size? Or was there a targeted size? Why cut the leaves into smaller fragments? How much leaf litter material (mass) was added to each bottle? How much water (1L?)

The leaf fragments were cut to a target size of 1.5 cm x 1.5 cm. The leaves were cut to allow measurements in smaller volumes of water. We do not have an estimate of the mass of leaves in each 1-L bottle. We added approximately 1 L of water to each bottle. These lines now read: "We removed a subset of the bags on 11 March 2021 and returned them to the laboratory, where we cut the leaves into fragments that were approximately 1.5cm x 1.5cm. We placed these fragments in 1-L bottles full of aerated stream water, which we incubated in water baths at five different temperatures (4, 8, 12, 16, 20ºC)." Page 4, Lines 105-107

Line 113: Were blanks measured initially and after the end of the incubation the same way? I worry about displacement/replacement of water due to the initial DO measurement given how large DO probes can be compared to a scint vial.

The blanks and samples were measured the same way before and after the experiment. Roughly 15% of the water in the vial was displaced during the measurement of the initial DO concentration, and it was immediately replaced with stream water from the same source that was used to fill the vial. We have added this information to the methods. These lines now read: "We used three subsamples from each replicate bottle to measure respiration rates. To estimate respiration rates, we filled 20-ml scintillation vials with stream water at the appropriate treatment temperature and measured the initial concentration of oxygen using a YSI 5100 Dissolved Oxygen Meter (YSI Inc, OH, USA). After measuring initial concentrations of oxygen, we immediately replaced the water that was displaced during the initial measurement (~15% of the volume) with stream water from the same bottle that was initially used to fill the vial." Page 4, Lines 112-116

Line 113: Was there at least an attempt to add a similar amount of leaf material to each vial?

Yes, we attempted to add a similar amount of leaf material to each vial. This line now reads: "Then, we added several leaf fragments (similar amounts among vials) to the vial and secured the cap such that no air remained in the vial". Page 4, Lines 116-117

Line 117: I suggest writing this out as an equation. Were the incubations done in the dark? I don't see anywhere suggesting that. If incubations were not done in the dark then this approach yields NEP, not respiration.

The incubations were done in the dark. This line now reads: "We then returned the vials to the water bath to incubate in the dark for 2 to 7 hours, giving the vials in colder temperatures more time to incubate to ensure meaningful changes in the concentration of dissolved oxygen. Page 4 " Lines 118-120 We now describe the math in more detail in the text and provide an equation. The text now reads: "After incubation, we recorded the final concentration of dissolved oxygen, removed the leaves, dried them to a constant mass, and weighed them. We calculated respiration rates (mg $O_2$ hr$^{-1}$ mg$^{-1}$) based on the difference in the mass of oxygen in the vial from before ($O_{2-pre}$, mg) and after the incubation ($O_{2-post}$, mg), minus the change in oxygen in the blanks ($O_{2-pre-blank}$ and $O_{2-post-blank}$ ), divided by incubation time (T, hr) and the dry mass of leaves in the vial (M, g, equation 1). $Respiration\ rate = \frac{(O_{2-pre}-O_{2-post})-(O_{2-pre-blank}-O_{2-post-blank})}{T*M}$ (1)" Page 4-5 Lines 120-125

Line 123: nominal pore size?

The nominal pore size was 1.0 micron. This line now reads "After 2-7 hours of incubation, we removed a subsample of the water with a syringe and filtered it through an AE-grade glass fiber filter (nominal pore size 1.0-μm, Sterilitech, WA, USA), and immediately froze the sample" Page 5, Lines 130-132.

Line 126: U is traditionally reported in units of mass per area per time (e.g., mg P / m2 / h). I think the approach the authors have taken here to estimate uptake as mass of nutrient per mass of leaf per time is fine but I think that something other than just "U" should be used here. Also, I'm not the biggest fan of the calculation for U as I would greatly prefer an initial and final sample collected from the same sample container. The authors are assuming that all incubation vials started with the same conditions. I don't know how I feel about that assumption. Were individual tubes amended with P? Or was a reservoir amended with P and then added to the tubes? Also, the drastic differences in incubation time is strongly suggesting an assumption of linear P uptake which I don't know can be expected to hold true across different concentrations.

For the sake of clarity, we would like to use $U$, which we define clearly at first usage. For each temperature treatment a reservoir of stream water was amended with P, shaken to mix the solution, and then this nutrient-amended water was added to the vials that were incubated. The text describing this now reads: "We amended reservoirs of stream water at each temperature with nutrients to elevate concentrations from <5 μg L$^{-1}$ to ~30 to 60 μg L$^{-1}$ SRP. We then dispensed 40 mL of this nutrient-amended water into 50-mL centrifuge tubes, and added several leaf fragments." Page 5 Lines 127-129. Our analysis does assume that all vials within a given temperature treatment had the same initial concentration. The three blanks that were measured

from each temperature treatment generally had measurement of P concentration that were within 1-2 µg L$^{-1}$ of one another, which gives us confidence that the reservoir was well mixed, and that the initial conditions were similar among vials. The approach we used allowed us to forgo removing water to measure P concentrations at the beginning of the experiment. When designing this experiment, we were concerned that taking an initial sample from each vial would change water volumes, which would change the total mass of P in the vial, and could have a large effect on rates given the small volumes of water we used. We used shorter incubation times for the warmer incubations in this experiment because we expected uptake and respiration to proceed more quickly. Our goal was to have similar masses of P taken up across the different temperature treatments to limit the influence of kinetic effects.

Line 135: Where was this categorical block effect included? What are the different experimental batches? I don't understand this statement at all.

We ran the experiment twice to increase our sample size. There may have been differences in the microbial communities, the stream water we collected, or other factors between the two batches of the experiment we ran, and we wanted to account for these in our statistical models. The block effect was included in the statistical model by adding a categorical variable that indicated which batch each measurement was taken from. In practice, this meant that the model estimated a different *y*-intercept for the data from each batch, and a single slope. This line now reads: "The two dates on which the experiment was run may have had different biological or environmental conditions, so we included a categorical effect of date in our statistical models to account for any differences" Page 6, Lines 146-147

Lines 136 – 138: What? I do not understand this statement at all. There was a model that compared respiration and Usrp to each other and a categorical variable that indicated if the model was for respiration or Usrp? What kind of a model? How was it evaluated? What was the dependent variable? The dependent variables? More detail and description is needed here.

The model compared the slopes of the relationship between temperature and each rate (respiration and uptake), which are the activation energies. The model is essentially an ANCOVA, but with temperature transformed so that slopes represent activation energies. Because the question we are testing is whether the slopes are different, we are interested in evaluating the significance of the interaction term in this model. These lines now read: "To evaluate whether the responses of respiration and U$_{srp}$ to temperature were different, we used an ANCOVA-type linear model. To do this, we fit a linear model that described the natural log$_e$-transformed rates of respiration and uptake rates as a function of the standardized Boltzmann temperature. The model included an interaction between temperature and a binary variable that indicated the type of rate (i.e., respiration or uptake). A significant interaction term in this model indicates that the slopes of the relationships between temperature and these rates differ." Page 6, Lines 147-152

Line 138: A significant interaction term between what?

The interaction term in the model is between temperature and a categorical variable that describes whether the rate is respiration or uptake (see revised text above).

Line 143: Why was the centered inverse Boltzmann temperature used as the predictor variable? Why not just temperature?

By regressing natural log-transformed rates of biological processes against the inverse Boltzmann temperature the slopes of these relationships can be interpreted biologically as an activation energy. We have added more description of this to the revised version of the manuscript. The revised text reads: "To quantify the effect of temperature on rates of respiration and $U_{srp,}$ we estimated their activation energies ($E_a$) using the Boltzmann-Arrhenius equation (equation 2, Brown et al., 2004), where the rate of the process ($r_i$) is a function of the rate at a reference temperature ($r_{ref}$), the activation energy ($E_a$), the temperature in kelvin ($T$), and the Boltzmann constant ($k_B$; $8.617\times10^{-5}$ eV K$^{-1}$)" Page 4-5, Lines 137-140

Line 149: Were leaves weighed at the end of the experiment tin the same manner? How were initial SRP concentrations achieved?

These methods were the same as those we described above for the first experiment. The leaves were weighed and the P concentrations were achieved by adding a concentrated solution of phosphorus to stream water. We have added the following text: "Leaf fragments incubated in each bottle were dried and weighed after the incubations." Page 6, Line 167-168 and "Leaves were incubated at six temperatures ranging from 4°C to 21°C and eight initial SRP concentrations ranging from 11 to 217 µg L-1 that were created by adding a concentrated solution of KH2PO4 to the stream water" Page 6, Lines 163-164

Lines 150 – 154: So 6 temperatures * 8 SRP concentrations * 3 bottles per treatment = 144 individual incubations. Is that accurate?

Yes, that is accurate.

Lines 145 – 173: So basically Usrp was regressed against temperature for each initial SRP concentration and then the slope of those regressions were compared across initial SRP concentration? Was a regression or correlation or something done here? There don't seem to be any stats, it reads like the authors plotted these out and visualized them but that's not a real satisfying analysis in my opinion. The same general though holds for the M-M analysis, too.

The regressions that we did in the section were not just visual - we fit regression models to the data. We have added these parameters as Tables A1 and A2 to the supplemental material.

Line 199: It seems like there should to be an analysis section. Or more detail needs to be given for the analyses in the individual sections (as was described in some of my previous comments). How were the simulation models assessed/evaluated?

Because there are three distinct units (two experiments and the simulations), we prefer to include the relevant analysis within each section, instead of describing all the analyses in one unified section. We evaluated the outcomes of the simulations by simply comparing the estimated effects. We did not include statistics for the simulations. We have clarified this in the text: "We propagate uncertainty in our parameter estimates of temperature sensitivities by

bootstrapping our estimates of cumulative $U_{srp}$ 1000 times, and compare outcomes of the simulations to estimate effect sizes. We do not include statistical analysis of the outcomes of the simulations." Page 8,  Lines 210-212

Line 203: canonical is an odd word choice here.

In this context, "canonical" means "according to recognized rules or scientific laws" and is conventionally used in the metabolic theory literature to describe the established value for cellular respiration's temperature dependence. This line now reads; "We estimated an $E_a$ of respiration during the laboratory experiment of 1.02 eV (SE 0.06), which is higher than the established canonical value for cellular respiration (0.60 - 0.70 eV, Figure 1a)." Page 8, Lines 216-217

Line 226: These are interesting results. It's definitely a very simplified model, but I think that is acknowledged and it points towards interesting (and testable) mechanisms changing P dynamics with future warming expectations. Obviously there are many other things to consider (e.g., changes in animal behaviors altering the decomposition of leaves, shifts in phenology matching shifts in climate, changes in N dynamics and broader stoichiometric questions…) but still an interesting exercise.

Thank you, we agree that our approach is simplified in terms of parts of the ecosystem that we model. We appreciate that other studies could add additional components, such as animal behavior, phenology, and broader stoichiometric issues.

Line 256: Again, I wonder how much temperature is correlated with canopy cover in some of these whole system nutrient spiraling studies. I also think it's difficult to compare a scint vial's worth of Usrp to a full stream nutrient release. The authors have quantified the effect of temperature on leaf respiration and leaf-based Usrp. They did not measure anything about other components of the ecosystem that could/would change with temperature (e.g., sediment uptake dynamics, hyporheic processes, autotrophic uptake (which would increase with decreasing temps due to autocorrelation with canopy cover). While I think the simulation study was a valuable exercise, I don't think that these results can really be extrapolated and compared to reach-scale results/studies.

We have added an acknowledgement of the potential role for autotrophic nutrient uptake (and other processes) to this section of the manuscript. This line now reads; "While these studies potentially illustrate the role of heterotrophic microbial biomass in nutrient uptake, the observed winter peaks in nutrient uptake in these studies may be driven in part by increased autotrophic production allowed by a relatively open canopy during winter" Page 10, Lines 277-279. While we don't contend that the rates we measure are equivalent to rates measured at the whole-stream scale, we do think it is useful to compare the uptake measurements we made in the laboratory with uptake measurements made at the stream-reach scale.

Lines 262 – 263: This statement is unfounded. I disagree that the current study separated the contribution of physiological and biomass-mediated effects of temperature on ecosystem-level nutrient uptake. As mentioned in the previous comment, the study separated the contribution of these temperature-based mechanisms to affect leaf litter respiration and Usrp. Even in the most

detritally-driven ecosystems (of which, Coweeta stream are definitely up near the top), there are still a multitude of other autotrohic and heterotrophic compartments contributing to ecosystem-scale respiration and nutrient uptake. This section should be either deleted or modified to be more accurate for what the study actually did do (which is still a valiant effort!).

We appreciate this point and have changed how we refer to the results of the simulations throughout the paper. Specifically, we no longer refer to the results of the simulations as describing "ecosystem nutrient uptake," but instead refer to the results as describing "long-term cumulative uptake". We hope that this revision makes the stated scope of our findings more in line with our analysis. The text now reads: "However, our study is the first to separate the contribution of these two processes to patterns of long-term cumulative uptake. Specifically, when we consider only the direct effect of temperature on mass-specific rates of $U_{srp}$, we infer that $U_{srp}$ increases with temperature. However, when we incorporated indirect effects of temperature on respiration and its consequences for biomass, we find that warming decreases cumulative $U_{srp}$ (Figure 4a)." Page 10, Lines 281-284

Line 265: But this ignores potential increases in sediment-based Usrp. Or autotrophic. Or hyporheric. Maybe the leaf litter breakdown is fueling more labile DOM to reach interstitial spaces where sorbed P can be broken down.

We agree that our model does not include these factors. We hope the revisions we have made above clarify our results.

Line 295: These are great caveats to include. I don't know how the authors can make the bold claims such as in lines 262 – 263 and then simultaneously acknowledge all of these issues.

We hope that the revised language we use above addresses these concerns.

Line 305:Not sure how this study addresses/provides insight into the effect of observational scale on temperature sensitivity.

The comparison we are looking at drawing here is between the measured instantaneous effects and the simulated long-term effects. In revising this manuscript, we have changed this from "observational scale" to "time scale". This line now reads: "The results of our study, although only a small step, highlight that nutrient uptake is dependent on temperature but uncoupled from increases in carbon demand, and the direction of the effect of warming on nutrient uptake is sensitive to the time scale that is considered (i.e., instantaneous vs. over months or years)." Page 11, Lines 324-326

Line 305: The majority (entirety?) of the discussion focuses on the simulation experiment, which is the weakest part of this paper in my opinion. I think more general discussion of temperature dependence of biogeochemical processes and how that can affect things more broadly would worth including initially, and then a toned down version of the focus on the simulation model. E.g. 'The results of our lab incubations would theoretically imply xyz. Our simulation studies confirm some of these expectations but revealed somewhat contradictory patterns due to abc.'

We hope that the revisions we have made through the discussion better contextualize this study.

Figure 1: I know the stats and fits are included in table 1, but I still think they'd be good to include on the figures, personally. I recommend including the equations for each line as well as stats (r-2, p-value).

We have added $R^2$ values to the figure captions, along with the slopes presented as activation energies.

Figure 2a: Are each of these significant regressions in panel a? Doesn't seem like it, which would entail no relationship between SRP uptake and temperature at certain initial values. A table (supplemental?) with slopes, r-squared, etc supporting these model fits would be useful. I'm not sure what in table 1 is showing model fit for these lines. Maybe a more details statistical analysis section in the methods would help clarify things a bit.

We have added appendix Table A1 that includes the model fits for Figure 2a.

Figure 2b: It almost looks like this is a hump-shaped relationship maxing out at mid-concentration. Inhibitory effect of high P? Any reason this particular form of curve was used?

If there was an inhibitory effect of high P, then we would expect lower rates of P uptake at all temperatures in the highest P concentration. We don't really see this in Figure 2a, where we visualize the rates directly. In Figure 2b we see that the response of P uptake to temperature does peak at intermediate concentrations, but we don't think that inhibition at high P can explain this. We used this fit because saturating functions are often used to describe the effect of increasing concentration on ecological processes.

Figure 3: Now by the third figure, I'm really having difficulty connecting individual panels from figures 1 – 3 to stats and model fits from table 1. Make it easy for me (and the other readers) by putting this info on the individual panels.

We have added $R^2$ values to panels B and C in figure 3, and reference appendix Table 2 in the caption for Figure 3, which has the fits for each line in panel A.